# Exposure to high-sugar diet induces transgenerational changes in sweet sensitivity and feeding behavior via H3K27me3 reprogramming

Jie Yang[1], Ruijun Tang[2], Shiye Chen[1], Yinan Chen[1], Kai Yuan[2,3]*, Rui Huang[4,5]*, Liming Wang[5]*

[1]Life Sciences Institute, Zhejiang University, Hangzhou, China; [2]Hunan Key Laboratory of Molecular Precision Medicine, Department of Neurosurgery, Xiangya Hospital, and Hunan Key Laboratory of Medical Genetics, School of Life Sciences, Central South University, Changsha, China; [3]The Biobank of Xiangya Hospital, Xiangya Hospital, Central South University, Changsha, China; [4]Center for Neurointelligence, School of Medicine, Chongqing University, Chongqing, China; [5]Institute of Molecular Physiology, Shenzhen Bay Laboratory, Shenzhen, China

*For correspondence:
yuankai@csu.edu.cn (KY);
huangrui85@cqu.edu.cn (RH);
lmwang83@szbl.ac.cn (LW)

Competing interest: The authors declare that no competing interests exist.

**Abstract** Human health is facing a host of new threats linked to unbalanced diets, including high-sugar diet (HSD), which contributes to the development of both metabolic and behavioral disorders. Studies have shown that diet-induced metabolic dysfunctions can be transmitted to multiple generations of offspring and exert long-lasting health burden. Meanwhile, whether and how diet-induced behavioral abnormalities can be transmitted to the offspring remains largely unclear. Here, we showed that ancestral HSD exposure suppressed sweet sensitivity and feeding behavior in the offspring in *Drosophila*. These behavioral deficits were transmitted through the maternal germline and companied by the enhancement of H3K27me3 modifications. PCL-PRC2 complex, a major driver of H3K27 trimethylation, was upregulated by ancestral HSD exposure, and disrupting its activity eliminated the transgenerational inheritance of sweet sensitivity and feeding behavior deficits. Elevated H3K27me3 inhibited the expression of a transcriptional factor Cad and suppressed sweet sensitivity of the sweet-sensing gustatory neurons, reshaping the sweet perception and feeding behavior of the offspring. Taken together, we uncovered a novel molecular mechanism underlying behavioral abnormalities spanning multiple generations of offspring upon ancestral HSD exposure, which would contribute to the further understanding of long-term health risk of unbalanced diet.

## Editor's evaluation

This study presents an important finding that high-sugar diet-induced behavioral changes can be transmitted to the offspring through the maternal germline. Using genetic and molecular biology approaches in the fruit fly model, the authors convincingly show that HSD has a transgenerational effect on PER, and that mothers fed an HSD produce progeny with globally elevated H3K27me3. The work will be of great interest to behaviorists and epigeneticists.

## Introduction

Dietary factors play a critical role in regulating multiple biological processes and influencing animal metabolism and behavior. For example, dietary restriction extends lifespan through metabolic regulation (*Anson et al., 2003*; *Wu et al., 2019*), while high-fat diet (HFD) and high-sugar diet (HSD) lead

to obesity and various metabolic dysfunctions (*Birse et al., 2010*; *Buettner et al., 2007*; *Palanker Musselman et al., 2011*). Evidence has also emerged indicating that dietary factors impact gene expression through epigenetic modifications, which may contribute to these metabolic syndromes (*Park et al., 2017*). In addition to the direct effects of dietary changes within the same generation of animals, dietary changes may also lead to alterations in the germline cells which exert long-lasting effects in the following generations. Studies on individuals who were born during the Dutch and Chinese famine demonstrate that prenatal exposure to undernutrition environments causes over-weight and insulin resistance in the offspring (*Heijmans et al., 2008*; *Li and Lumey, 2017*; *Ravelli et al., 1998*; *Tobi et al., 2014*). Animal models such as worms, flies, and mice also indicate that expo-sure to abnormal diets induces various transgenerational metabolic disorders, including diabetes, obesity, hyperlipidemia, and so forth (*Dunn and Bale, 2011*; *Huypens et al., 2016*; *Somer and Thummel, 2014*; *Stegemann and Buchner, 2015*; *Wei et al., 2014*).

Such transgenerational inheritance upon dietary changes is thought to be mediated by several epigenetic factors, including DNA methylation, non-coding RNA (ncRNA), and histone modifications (*Bohacek and Mansuy, 2015*; *Heard and Martienssen, 2014*; *Miska and Ferguson-Smith, 2016*; *Skvortsova et al., 2018*). For example, studies in *Caenorhabditis elegans* demonstrate that starvation alters organismal metabolism spanning three subsequent generations via small RNAs (*Rechavi et al., 2014*), and HFD induces lipid accumulation signals which can be transmitted to multiple generations through H3K4me3 modifications (*Wan et al., 2022*). Similarly, previous reports on HFD mouse model have shown that in utero exposure to HFD causes a metabolic syndrome through epigenetic modifi-cations of adiponectin and leptin signaling, and that sperm tsRNA signaling contributes to intergener-ational inheritance of an acquired metabolic disorder (*Chen et al., 2016*; *Masuyama and Hiramatsu, 2012*). Dietary factors can also alter animal behaviors. For example, HFD affects the feeding and cognitive behaviors of mice (*Arnold et al., 2014*; *Pendergast et al., 2013*). In human studies, children of famine survivors had higher chances to develop psychological trauma or insanity (*Kelly, 2019*; *Li et al., 2015*; *Painter et al., 2006*), which implies that an abnormal diet may lead to behavioral disor-ders with transgenerational inheritance.

The fruit flies *Drosophila melanogaster* is a valuable model for studying the transgenerational inher-itance of animal behaviors. The effects of diet changes on various fly behaviors have been demon-strated in flies. For example, starvation increases files' locomotion and food-seeking behavior (*Yu et al., 2016*), and HSD reshapes sweet perception and promotes feeding (*May et al., 2019*). More-over, there is accumulating evidence of diet-induced transgenerational inheritance in *Drosophila*. For example, changes in dietary yeast concentrations induce transgenerational somatic rDNA instability and copy number reduction (*Aldrich and Maggert, 2015*). HFD exposure induces transgenerational cardiac lipotoxicity through H3K27me3 modifications (*Guida et al., 2019*). A low protein diet leads to transgenerational reprogramming of lifespan through E(z)-mediated H3K27me3 modifications (*Xia et al., 2016*). There is also evidence that behavioral changes in *Drosophila* can be transmitted to subsequent generations: exposure to predatory wasps leads to transgenerational ethanol preference via maternal NPF repression (*Bozler et al., 2019*).

HSD results in many physiological responses and metabolic/behavioral disorders in the same gener-ation of *Drosophila* (*Chen et al., 2021*; *Chng et al., 2017*; *May et al., 2019*; *van Dam et al., 2020*). Some metabolic changes, such as obese- and diabetes-like phenotypes, can be passed on to their offspring through germline epigenetic alterations (*Buescher et al., 2013*; *Karunakar et al., 2019*; *Öst et al., 2014*; *Palanker Musselman et al., 2011*). However, the possibility and mechanisms of trans-generational inheritance of behavioral changes upon HSD exposure are far less studied in fruit flies.

In this study, we found that HSD induced metabolic and behavioral dysfunctions as previously reported, and discovered that HSD suppressed sweet sensitivity and feeding behavior in the offspring. Furthermore, chromatin-immunoprecipitation followed by sequencing (ChIP-seq) data revealed that this transgenerational behavioral change was mediated by upregulated H3K27me3 modifications transmitted through the maternal germline. More specifically, we identified that ancestral HSD expo-sure elevated H3K27me3 levels in the promoter region of *cad* gene, resulting in a reduction in its mRNA expression in the sweet-sensing gustatory neurons of offspring, eventually reshaping the sweet perception and feeding behavior. Taken together, our study uncovered a novel molecular mechanism underlying the transgenerational behavioral changes upon ancestral HSD exposure, and shed light on the understanding of long-term health risks of dietary abnormalities in human.

## Results

### HSD feeding suppressed sweet sensitivity and feeding behavior spanning multiple generations

Previous work has shown that ancestral exposure to abnormal diets (such as HFD and HSD) led to various metabolic dysfunctions in the offspring, including cardiac lipotoxicity, diabetes, and obesity (*Chen et al., 2022*; *Guida et al., 2019*; *Kaspar et al., 2020*; *Wan et al., 2022*). However, whether ancestral experience exerted transgenerational behavioral modulations was still unclear. To address this question, we used *D. melanogaster* as a model system to examine the potential transgenerational behavioral effect of HSD.

Wild-type flies were raised with HSD from the embryo stage to adulthood (termed HSD-F0 flies). Fresh embryos of HSD-F0 flies were transferred back to normal diet (ND) and raised on ND until adulthood (termed HSD-F1 flies). These HSD-F1 flies were further raised and mated on ND to generate multiple generations of offspring (HSD-F2 to -F5 flies). Flies continuously raised on ND without any HSD exposure were used as ND-fed controls (*Figure 1A*). Essentially, HSD-F1 to -F5 flies and ND-fed control flies were all raised on ND food from their embryo stage, thus any metabolic and behavioral differences among them were likely attributed to ancestral exposure to HSD and its transgenerational effect on offspring.

We first assayed multiple physiological and metabolic parameters of HSD-F0 flies versus ND-fed controls to validate our HSD-feeding protocol. Compared to ND flies, HSD-F0 flies exhibited decreased body weight (*Figure 1B*). In contrast, their triglyceride and glycogen storage levels, as well as trehalose levels, the major circulating sugar in the fly hemolymph, were elevated (*Figure 1—figure supplement 1A–C*). Hyperglycemia together with weight loss was a sign of insulin signaling dysfunction. We therefore quantified the mRNA levels of two important insulin-like molecules in flies, *Drosophila* insulin-like peptide 2 (DILP2) and DILP5, which were both released by insulin-producing cells in the fly brain upon nutrient uptake (*Ikeya et al., 2002*). As expected, the expression levels of these genes were downregulated in HSD-F0 flies (*Figure 1—figure supplement 1D–E*).

These data suggest that HSD feeding leads to the development of a diabetes-like phenotype in the same generation (HSD-F0) as previously reported (*Chen et al., 2021*; *Palanker Musselman et al., 2011*; *van Dam et al., 2020*). Furthermore, we also found out that HSD-F1 flies exhibited similar physiological and metabolic changes as HSD-F0 flies despite the former not experiencing HSD feeding (*Figure 1C*, *Figure 1—figure supplement 1A–E*), suggesting transgenerational inheritance of metabolic programming upon ancestral HSD exposure (*Buescher et al., 2013*).

Next, we asked whether ancestral HSD exposure resulted in behavioral abnormalities and whether this effect could also be transmitted to the offspring. Given that insulin signaling played an important role in feeding regulation (*Porte et al., 2002*), we first measured total food consumption by the Capillary Feeder (CAFE) assay in HSD-F0 flies (*Ja et al., 2007*). As previously reported, HSD-F0 flies exhibited significantly increased food consumption compared to ND-fed controls in a 24 hr duration (*Figure 1—figure supplement 2A*); similar increases were observed in HSD-F1 and HSD-F2 flies, too (*Figure 1—figure supplement 2B*). However, when we used our previously developed Manual Feeding (MAFE) assay (*Qi et al., 2015*) to measure the volume of ingested food by individual flies during the course of a single meal, we found that HSD-F0 flies exhibited significantly decreased food consumption (*Figure 1D*), and that the suppression of meal size was transmitted through multiple generations (*Figure 1E*).

A major difference between these two feeding assays was that in the MAFE assay flies were immobilized and presented with microcapillaries filled with liquid food, whereas in the CAFE assay flies were free moving and could decide when to feed. Therefore, a major determinant of the readout of the MAFE assay was whether flies were responsive to the presented food and were willing to extend their proboscis to initiate a meal, and that of the CAFE assay was flies' overall energy need. Thus, the discrepancy between the results from the CAFE assay and the MAFE assay suggests that upon HSD exposure, flies' overall energy expenditure is elevated while their sweet perception is inhibited, hence their increased food consumption in the CAFE assay but reduced meal size in the MAFE assay.

To test this hypothesis, we used proboscis extension reflex (PER), a behavioral component of feeding initiation (*Inagaki et al., 2012*), to examine sweet sensitivity of these flies (*Figure 1F*). We found that both starved and fed HSD-F0 flies showed reduced PER responses to various concentrations of sucrose compared to ND-fed controls (*Figure 1G*, *Figure 1—figure supplement 2C*), and

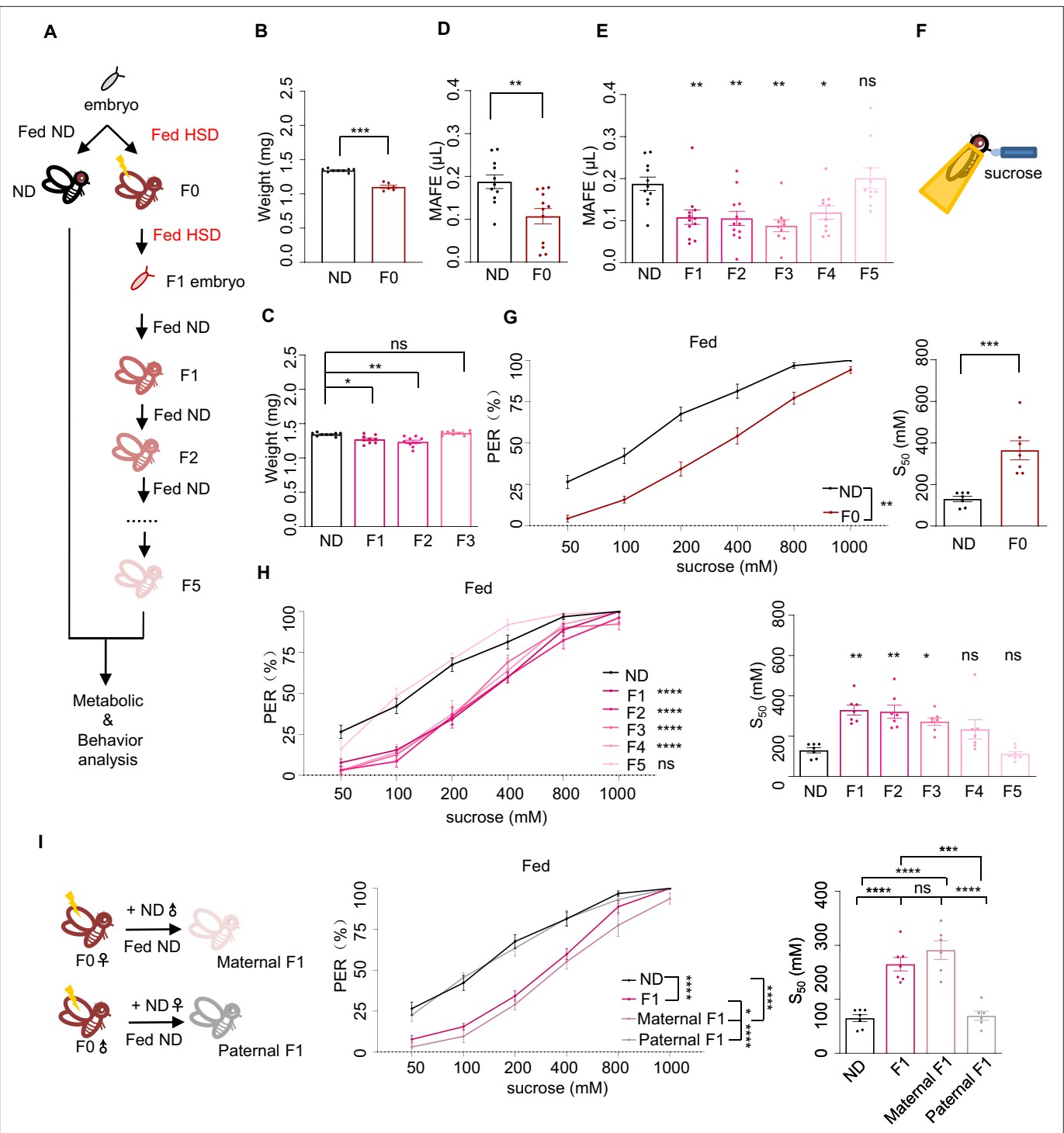

**Figure 1.** Ancestral high-sugar diet (HSD) exposure decreased sweet sensitivity and feeding behavior across multiple generations of offspring. (**A**) The illustration of experimental design for **B**-**I**. The embryos of wild-type *Canton-S* flies were collected and fed with normal diet (ND) (black, referred to ND controls) or HSD (red, referred to HSD-F0) until maturity. HSD-F0 flies were mated to produce the next generation (HSD-F1). The embryos of HSD-F1 flies were transferred to ND right after egg laying and kept on ND until adulthood. HSD-F1 flies were mated to propagate multiple generations of offspring (HSD-F2 to F5) on ND diet for metabolic and behavioral assays. (**B–C**) The body weight of individual flies from different treatment groups (n=6 biological replicates, each containing 5 flies). (**D–E**) Volume of 400 mM sucrose consumed by individual flies using the Manual Feeding (MAFE) assay (n=10–12). (**F**) Schematic illustration of the proboscis extension reflex (PER) assay. (**G–I**) Fractions of flies showing PER responses to different concentrations of sucrose (n≥6 biological replicates, each containing 8–12 flies). The $S_{50}$ indicated the sucrose concentration that induced PER responses in 50% of the tested flies. Data were shown as means ± SEM. ns p>0.05; *p<0.05; **p < 0.01; ***p < 0.001; ****p < 0.0001.

*Figure 1 continued on next page*

*Figure 1 continued*

The online version of this article includes the following source data and figure supplement(s) for figure 1:

**Source data 1.** Raw data of the metabolic and behavioral experiments shown in *Figure 1*.

**Figure supplement 1.** Ancestral high-sugar diet (HSD) exposure induced metabolic changes in the offspring.

**Figure supplement 1—source data 1.** Raw data of the metabolic and qPCR experiments shown in *Figure 1—figure supplement 1*.

**Figure supplement 2.** Ancestral high-sugar diet (HSD) exposure induced behavioral changes in the offspring.

**Figure supplement 2—source data 1.** Raw data of the metabolic and behavioral experiments shown in *Figure 1—figure supplement 2*.

such effect could transmit to following generations till HSD-F4 (*Figure 1H*, *Figure 1—figure supplement 2D*).

It was worth noting that the effect size of acute HSD exposure (in HSD-F0 flies) was stronger than that in the offspring (in HSD-F1 flies and their progeny), which was as expected and with clear biological relevance (*Bozler et al., 2019*; *Klosin et al., 2017*). In all living organisms, rapid changes in the environment, including alterations in diet, exert immediate and profound impacts on their survival and reproduction, hence requiring prompt and robust responses. In contrast, clues from ancestral experience may only offer vague and indirect clues of the current living conditions of the offspring. While such information may still benefit the survival and reproduction of the offspring, its effect size is expected to become smaller through multiple rounds of reproduction.

Since HSD exposure modulated both metabolism and feeding behavior in the progeny, it was possible that these two effects were connected, that is altered metabolism upon ancestral HSD exposure affected feeding behavior. However, there were several lines of evidence suggesting against this possibility. For example, the body weight and nutrient storage (triglyceride, glycogen, and circulating trehalose) returned to normal in HSD-F2 or HSD-F3 flies (*Figure 1C*, *Figure 1—figure supplement 1A–C*), whereas HSD-F2 to HSD-F3 flies still exhibited reduced meal size (*Figure 1E*) and PER responses to sucrose (*Figure 1H*, *Figure 1—figure supplement 2D*). In addition, DILP2 and DILP5 expression levels were both reduced by HSD exposure from HSD-F0 to HSD-F2 flies but not in HSD-F3 flies (*Figure 1—figure supplement 1D–E*; *Nässel et al., 2013*). Therefore, it was unlikely that altered metabolism or insulin signaling was the causal factor for the reduction in sweet sensitivity and feeding behavior in HSD-exposed flies.

We also asked whether HSD exposure specifically modulated sweet sensitivity or the gustatory system in general. We found that both HSD-F0 and HSD-F1 flies exhibited similar gustatory responses to fatty acid (1% hexanoic acid), another type of appetitive stimuli (*Ahn et al., 2017*; *Brown et al., 2021*), compared to ND-fed controls (*Figure 1—figure supplement 2E–F*). Therefore, it is likely that HSD exposure specifically modulates sweet sensitivity in a transgenerational manner.

Notably, we chose to use female flies throughout the current study since their behavioral measures were more stable than males. But we verified that HSD exposure also suppressed sweet sensitivity in males, both in HSD-F0 and HSD-F1 flies (*Figure 1—figure supplement 2G–H*). Therefore, transgenerational inheritance could be mediated by the male or female parent as shown in previous work (*Daxinger and Whitelaw, 2012*; *Heard and Martienssen, 2014*). To distinguish between these two possibilities, we performed HSD feeding in only female or male F0 flies and crossed them with ND-fed mates. As shown in *Figure 1I*, the reduction in PER was only seen in F1 flies with HSD-fed female ancestor but not with HSD-fed male ancestor. Similar results were observed in the F2 generation, that only F2 flies with HSD-F1 female ancestor exhibited reduction in PER (*Figure 1—figure supplement 2I*). These results suggest that the effect of HSD exposure is transmitted to offspring via female gametes.

## Ancestral HSD exposure elevated genome-wide H3K27me3 levels in offspring

Next, we investigated the underlying mechanism of transgenerational behavioral inheritance after ancestral HSD exposure. We focused on epigenetic regulators since it was unlikely that HSD exposure resulted in specific genetic alterations in the germline cells of ancestral flies. Despite the importance of DNA methylation in the regulation of vertebrate transgenerational inheritance, it was reported that DNA methylation in *Drosophila* was negligible and limited to the early stages of embryonic development (*Lyko et al., 2000*). Multiple lines of research indicated that two types of histone methylations,

H3K27me3 and H3K9me3, played important roles in transgenerational inheritance in *Drosophila* (*Coleman and Struhl, 2017*; *Wang and Moazed, 2017*). In addition, piwi-interacting RNA (piRNA), an important species of ncRNA in transgenerational inheritance of *Drosophila*, was associated with altered H3K27me3 and H3K9me3 (*Le Thomas et al., 2014*; *Peng et al., 2016*). Therefore, we asked whether ancestral HSD exposure induced alterations in post-translational modifications of H3K27 and H3K9.

We collected embryos at mitotic cycle 10–12 of both ND and HSD-F1 flies, and performed ChIP-seq using antibodies against four histone modifications (H3K27me3, H3K27ac, H3K9me2, and H3K9me3) respectively (*Figure 2A*). We performed peak calling and determined the occupancy of H3K27me3, H3K27ac, H3K9me2, H3K9me3, and H3 on the fly genome. Genomic snapshots of representative target loci (Hox cluster genes: *bxd*, *Ubx*, and *Abd-A*) confirmed H3K27me3, H3K27ac, and H3K9me3 enrichment as expected (*Figure 2B*, *Figure 2—figure supplement 1A*). Meanwhile, we also observed that H3K27me3 and H3K27ac were widely spread in the genome at this stage, while H3K9me3 was preferentially localized within gene desert regions and heterochromatic regions such as centromeres and pericentromeric (data not shown). Our analysis also confirmed that H3K9me2 was nearly undetectable in euchromatic regions during this stage (*Figure 2B*).

We then identified genomic regions that were significantly enriched for H3K27me3, H3K27ac, and H3K9me3 in HSD-F1 embryos using model-based analysis of ChIP-seq 2 (MACS2) and generated the average signal density spanning a 2 kb region at both ends. We found that upon ancestral HSD exposure, the average peak intensity of H3K27me3 modifications increased significantly (*Figure 2C*, *upper*) whereas the intensity of H3K27ac and H3K9me3 only exhibited modest increase (*Figure 2C*, *middle* and *lower*). In addition, previous work has also indicated that *Drosophila* oocytes transmitted repressive H3K27me3 marks to their offspring and exerted developmental impact (*Zenk et al., 2017*). Thus, we focused on the upregulation of H3K27me3 signals upon ancestral HSD exposure for further analysis.

To better understand the upregulation of H3K27me3 peaks in HSD-F1 embryos, we performed differential peaks analysis and found that ancestral HSD exposure resulted in approximately 400 regions with H3K27me3 upregulation ($\log_{10}$ likelihood ratio >3) distributed throughout the genome and 6 H3K27me3 downregulated regions (*Supplementary file 1*). The signal intensity of H3K27me3 increased remarkably in those upregulated regions (*Figure 2D*, *left*). In these regions, the signal intensity of H3K27ac was significantly decreased (*Figure 2D*, *middle*), in line with the antagonism between H3K27ac and H3K27me3 modifications (*Tie et al., 2009*). Meanwhile, H3K9me3 signal in these regions remained unchanged (*Figure 2D*, *right*).

To validate the ChIP-seq results, we performed western blot on cycle 10–12 embryos and confirmed that the H3K27me3 levels in HSD-F1 embryos were significantly increased compared to ND embryos (*Figure 2—figure supplement 1B*). In line with these results, later-stage HSD-F1 embryos (cycle 13 and 14) also exhibited increased H3K27me3 modifications as directly revealed by H3K27me3 antibody staining (*Figure 2—figure supplement 1C*). H3K27me3 modifications in the adult stage of HSD-F1 flies were also upregulated compared to ND (*Figure 2—figure supplement 1D*). We also found that HSD exposure enhanced H3K27me3 modifications in the whole body as well as in the ovary of HSD-F0 flies (*Figure 2—figure supplement 1E*), further indicating that H3K27me3 modifications are formed in the female germline of HSD-F0 flies and transmitted to the offspring.

We then sought to understand how such modifications were maintained in the offspring. Previous studies reported that Polycomb-like protein (Pcl) interacts with Polycomb repressive complex 2 (PRC2) to constitute a specific form of PCL-PRC2 complex, which generated high levels of H3K27me3 on specific genomic regions (*Nekrasov et al., 2007*). Besides Pcl, PRC2 complex was comprised of several major components, including enhancer of zeste (E(z)), suppressor of zeste 12 (Su(z)12), extra sex combs (Esc), and chromatin assembly factor-1 (Caf-1) (*Figure 2—figure supplement 2A*; *Margueron and Reinberg, 2011*). By using quantitative RT-PCR, we found that the mRNA expression of Pcl was upregulated in HSD-F1 and HSD-F2 flies compared to ND controls, while mRNA expression of E(z), Su(z)12, Esc, and Caf-1 only exhibited modest yet insignificant changes (*Figure 2—figure supplement 2B–C*). These results suggest that PCL-PRC2 complex may be involved in the maintenance of H3K27me3 modifications in the offspring upon ancestral HSD exposure. Paradoxically, Pcl and E(z) expressions were not changed in HSD-F0 flies (*Figure 2—figure supplement 2D–G*), suggesting that the influence of acute HSD exposure on flies is more complex. For example, we also

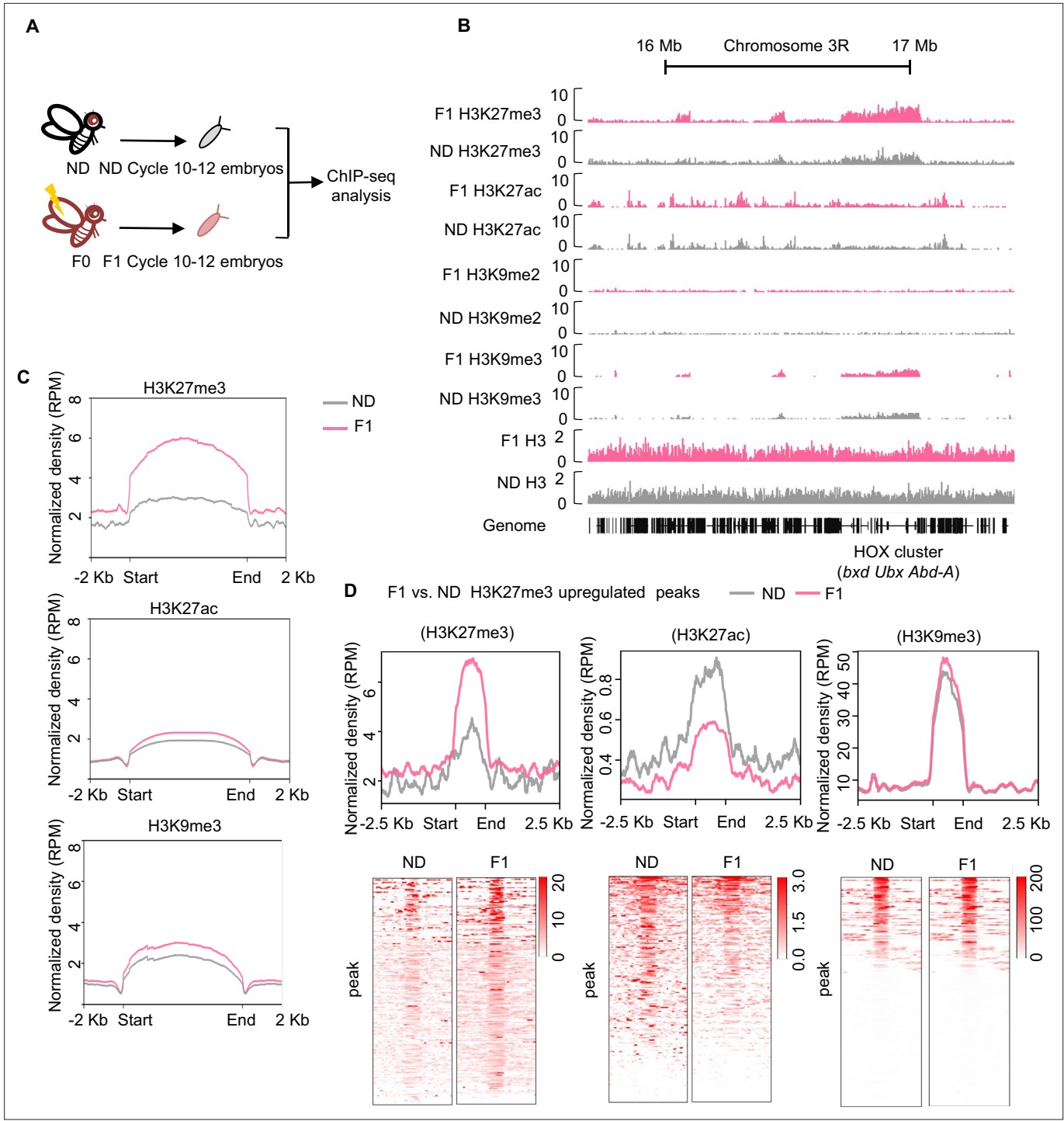

**Figure 2.** Ancestral high-sugar diet (HSD) exposure increased genome-wide H3K27me3 levels in the offspring. (**A**) The workflow of the chromatin-immunoprecipitation followed by sequencing (ChIP-seq) assay. Embryos of normal diet (ND) and HSD-F1 flies were collected from population cages at 25°C for 30 min, and allowed to develop for 80 min to target mitotic cycle 10–12 for ChIP-seq analysis. (**B**) Genome browser view of H3K27me3, H3K27ac, H3K9me2, H3K9me3, and H3 density at the HOX cluster (*bxd*, *Ubx*, and *Abd-A*) gene regions in embryos of ND and HSD-F1 embryos. (**C**) Average density plots showing the signal profiles of H3K27me3, H3K27ac, and H3K9me3 at their peaks plotted across a 4 kb window (±2 kb around the start/end of signals). (**D**) Average density plots (top) and heatmap (bottom) showing the distribution for the changes of H3K27me3, H3K27ac, and H3K9me3 signals for regions with upregulated H3K27me3 peaks in HSD-F1 embryos, respectively. Color bar showed the Z-score value in the heatmap.

The online version of this article includes the following source data and figure supplement(s) for figure 2:

**Figure supplement 1.** Ancestral high-sugar diet (HSD) exposure increased H3K27me3 levels in both maternal and offspring.

*Figure 2 continued on next page*

*Figure 2 continued*

**Figure supplement 1—source data 1.** Raw data of the western blot and immunofluorescence staining experiments shown in *Figure 2—figure supplement 1*.

**Figure supplement 2.** Ancestral high-sugar diet (HSD) exposure enhanced *Pcl* expression in the offspring.

**Figure supplement 2—source data 1.** Raw data of the qPCR experiments shown in *Figure 2—figure supplement 2*.

noted that the Utx histone demethylase (*Utx*) gene as well as CREB-binding protein (*CBP*, encoded by nej) gene expression were downregulated in HSD-F0 flies (*Figure 2—figure supplement 2H*). Utx is a H3K27me3 demethylase known to associate with the histone acetyltransferase CBP and to directly block H3K27 trimethylation by E(z) (*Tie et al., 2009*). Given that genomic histone modifications undergo dynamic reprogramming during embryonic development, how such histone imprinting transmits to multiple generations of offspring remains unclear and is of significance for future studies (*Heard and Martienssen, 2014*).

## H3K27me3 was required for the transgenerational modulation of sweet sensitivity and feeding behavior upon ancestral HSD exposure

Since our data showed that HSD exposure elevated genome-wide H3K27me3 modifications in HSD-F1 embryos, we asked whether this inherited epigenetic change contributed to transgenerational behavioral deficits. To test this, we knocked down the expression of H3K27me3 catalytic enzyme in the PCL-PRC2 complex, E(z), to reduce H3K27me3 imprinting during embryogenesis by using *nosNGT-GAL4* which was active during the blastoderm stage of embryogenesis (*Tracey et al., 2000*; *Figure 3—figure supplement 1A*). We observed that compared to transgenic controls, *nosNGT-GAL4>UAS-E(z) RNAi* flies exhibited similar PER responses to sucrose in ND vs. HSD-F1 flies, suggesting the loss of transgenerational behavioral deficits (*Figure 3—figure supplement 1B*). Meanwhile, we observed that *nosNGT-GAL4>UAS-E(z) RNAi* flies still exhibited reduced PER responses to sucrose in HSD-F0 vs. ND flies, suggesting that E(z) is not required for acute HSD exposure-induced PER reduction (*Figure 3—figure supplement 1C*). Knocking down Pcl, another component of PCL-PRC2 complex, generated similar effect as E(z) (*Figure 3—figure supplement 1D*). Both E(z) and Pcl RNAi treatments did not alter the sweet sensitivity of ND-fed flies across different genotypes (*Figure 3—figure supplement 1C,D*). In contrast, RNAi knockdown of histone deacetylase Rpd3 and H3K9me3 methylase Su(var)3–9 did not affect the transgenerational behavioral inheritance upon ancestral HSD exposure (*Figure 3—figure supplement 2A,B*). These data indicate that H3K27me3 modification plays a crucial role in the transgenerational inheritance of sweet sensitivity and feeding behavior upon ancestral HSD exposure.

We reasoned that H3K27me3 imprinting, the driver of the transgenerational inheritance of sweet taste deficits upon ancestral HSD exposure, was transmitted through the maternal germline. To directly test this hypothesis, we used an maternal germline-specific GAL4 driver, *maternal alpha-tubulin GAL4* (*Matα-tub-GAL4*), to knock down E(z) and Pcl expression during oogenesis (*Figure 3A* **Hudson and Cooley, 2014**). Knockdown of E(z) in female germline eliminated H3K27me3 modifications in oocytes and ovary, as demonstrated by western blot and immunofluorescence staining (*Figure 3B and C*). We found that knockdown of E(z) and Pcl in female germline eliminated the suppression of PER responses in HSD-F1 flies without interfering with PER responses in ND-fed flies (*Figure 3D and E*). Such treatments but didn't change PER responses of HSD-F0 toward sucrose (*Figure 3—figure supplement 3A, B*), either. These results confirm that the transmission of maternal H3K27me3 modifications is critical for the transgenerational inheritance of sweet sensitivity and feeding behavior.

## H3K27me3 regulated the sensitivity of sweet-sensing gustatory neurons

As we showed above, elevated H3K27me3 modifications upon ancestral HSD exposure mediated the transgenerational behavioral inheritance. We next examined whether modulating H3K27me3 levels could directly cause changes in sweet sensitivity and feeding behavior.

We tested the effect of EED226, a potent PRC2 inhibitor that directly binds to the H3K27me3 binding pocket and suppresses H3K27me3 modifications (*Figure 4—figure supplement 1A*; *Loh et al., 2021*). We fed EED226 to HSD-F1, HSD-F2 and HSD-F3 flies for five consecutive days and found their PER responses to sucrose could all be restored to the levels of ND controls (*Figure 4A*, *left*

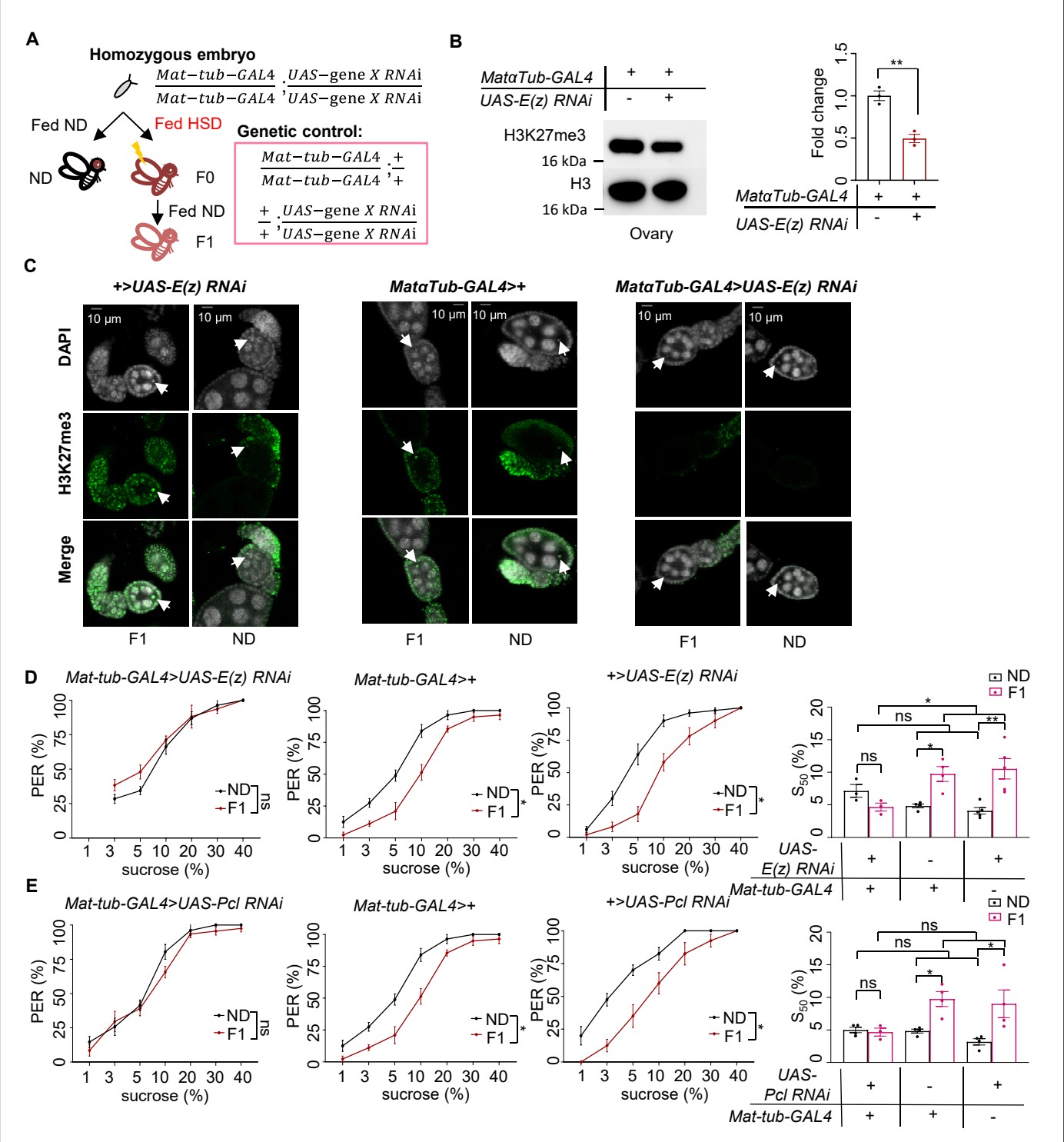

**Figure 3.** H3K27me3 modifications were essential for the transgenerational regulation of sweet sensitivity and feeding behavior upon ancestral high-sugar diet (HSD) exposure. (**A**) Schematic diagram of the PRC2 germline RNAi experiment. Note that *UAS-gene X RNAi* was integrated into both two third chromosomes and *Mat-tub-GAL4* was integrated into both two second chromosomes. The embryos of *Mat-tub-GAL4>UAS gene X RNAi* flies were collected and fed with normal diet (ND) (black, referred to ND controls) or HSD (red, referred to HSD-F0) until maturity. HSD-F0 flies were mated to produce the next generation (HSD-F1). These flies were prepared for biochemical and behavioral assays. (B) H3K27me3 modification levels in fly ovaries were analyzed by western blot (n=3 biological replicates, each containing 15 flies). Antibodies against H3K27me3 and H3 proteins were used in the western blot. (C) H3K27me3 modification levels in fly germarium were analyzed by immunofluorescence staining. Antibodies against H3K27me3 proteins and DAPI were used in the immunofluorescence staining. In the images H3K27me3 was shown in green and DAPI in white. Examples of oocyte nucleus

*Figure 3 continued on next page*

*Figure 3 continued*

were indicated by white arrowheads. Scale bar, 10 µm. (D–E) Fractions of flies of the indicated genotypes showing proboscis extension reflex (PER) responses to sucrose (n=3–5 biological replicates, each containing 8–12 flies). The $S_{50}$ indicated the sucrose concentration that elicited PER responses in 50% of the tested flies. Data were shown as means ± SEM. ns p>0.05; *p<0.05; **p<0.01; ***p<0.001; ****p<0.0001.

The online version of this article includes the following source data and figure supplement(s) for figure 3:

**Source data 1.** Raw data of the western blot, immunofluorescence staining, and behavioral experiments shown in *Figure 3*.

**Figure supplement 1.** H3K27me3 in the early embryo was required for transgenerational modulation of sweet sensitivity and feeding behavior.

**Figure supplement 1—source data 1.** Raw data of the behavioral experiments shown in *Figure 3—figure supplement 1*.

**Figure supplement 2.** H3K27ac and H3K9me3 in the early embryo were not required for transgenerational modulation of sweet sensitivity and feeding behavior.

**Figure supplement 2—source data 1.** Raw data of the behavioral experiments shown in *Figure 3—figure supplement 2*.

**Figure supplement 3.** H3K27me3 in the germline was not required for sweet sensitivity and feeding behavior induced by high-sugar diet (HSD) exposure.

**Figure supplement 3—source data 1.** Raw data of the behavioral experiments shown in *Figure 3—figure supplement 3*.

and *middle*). Chaetocin, a specific inhibitor of H3K9 methyltransferase Su(var)3–9 (*He et al., 2012*), did not rescue the sweet taste defects in these flies (*Figure 4A*, *right*). We further examined the effect of pharmacological inhibition of H3K27me3 by feeding HSD-F1 flies with either EED226 or A395, another histone methyltransferase inhibitor occupying the H3K27me3 binding sites (*He et al., 2017*), and found that both treatments restored sweet responses toward various concentrations of sucrose to the levels of ND-fed controls (*Figure 4B*). Meanwhile, pharmacological inhibition of H3K27me3 alleviated taste deficits in HSD-F0 flies, suggesting that elevated H3K27me3 modifications also partially mediates the effect of acute HSD exposure (*Figure 4C*).We also speculated that suppressing H3K27me3 modifications in adult flies would restore sweet taste sensitivity not only to themselves but also to their offspring. To test this, we fed EED226 to HSD-F1 flies and found restored PER responses in both HSD-F1 and their offspring, HSD-F2 flies (*Figure 4D*). These results phenocopied the effect of RNAi knockdown of E(z) and Pcl, further confirming the role of H3K27me3 modifications in mediating transgenerational behavioral heritance upon ancestral HSD exposure.

We next sought to understand the underlying neurobiological mechanism of transgenerational sweet taste deficits upon ancestral HSD exposure. Gustatory neurons expressing a gustatory receptor Gr5a played a central role in sugar perception (*Dahanukar et al., 2007*). We thus hypothesized that H3K27me3 modifications might modulate sweet sensitivity of Gr5a+ gustatory neurons. To directly test this, we knocked down *E(z)* in Gr5a+ gustatory neurons and indeed found a restoration of PER responses in both HSD-F0 and HSD-F1 flies (*Figure 4—figure supplement 1B*). To directly test the effect of H3K27me3 modifications on the activity of Gr5a+ neurons, we ectopically expressed a genetically encoded calcium indicator GCaMP6m in Gr5a+ neurons and conducted live calcium imaging during sucrose feeding episodes (*Yang et al., 2018*; *Figure 4E*, *left*). HSD-F1 flies exhibited significantly reduced calcium transients upon sucrose stimulation compared to ND flies, which could be restored by EED226 feeding (*Figure 4E*, *middle and right*, *Figure 4—figure supplement 1C*).

## Cad mediated the reduction in sweet sensitivity caused by ancestral HSD exposure

We then asked how elevated H3K27me3 modifications altered sweet sensitivity of Gr5a+ neurons. Previous work reported that the formation of H3K27me3 modifications maintained the silenced state of *Drosophila* homeobox genes and transmitted such a repressive state through multiple rounds of DNA replication in early embryos and exerted long-lasting impacts into adulthood (*Coleman and Struhl, 2017*; *Skvortsova et al., 2018*). We speculated that upregulated H3K27me3 in early embryos could alter gene transcriptome in certain organs and affect the physiology and behavior of adults.

To survey a broader range of candidate genes for further analysis, we used less stringent criteria compared to *Figure 2* ($log_{10}$ likelihood ratio >1) and identified ~3000 H3K27me3 hypermethylated regions in HSD-F1 flies using a window of ± 1 kb from the transcription start site and found 341 genes corresponding to hypermethylated sites (*Supplementary file 2*). Since H3K27me3 often marked the downregulation of gene expression, we then asked whether and which of these genes were transcriptionally suppressed. To achieve this, we performed RNA-seq analysis using head tissue of HSD-F1

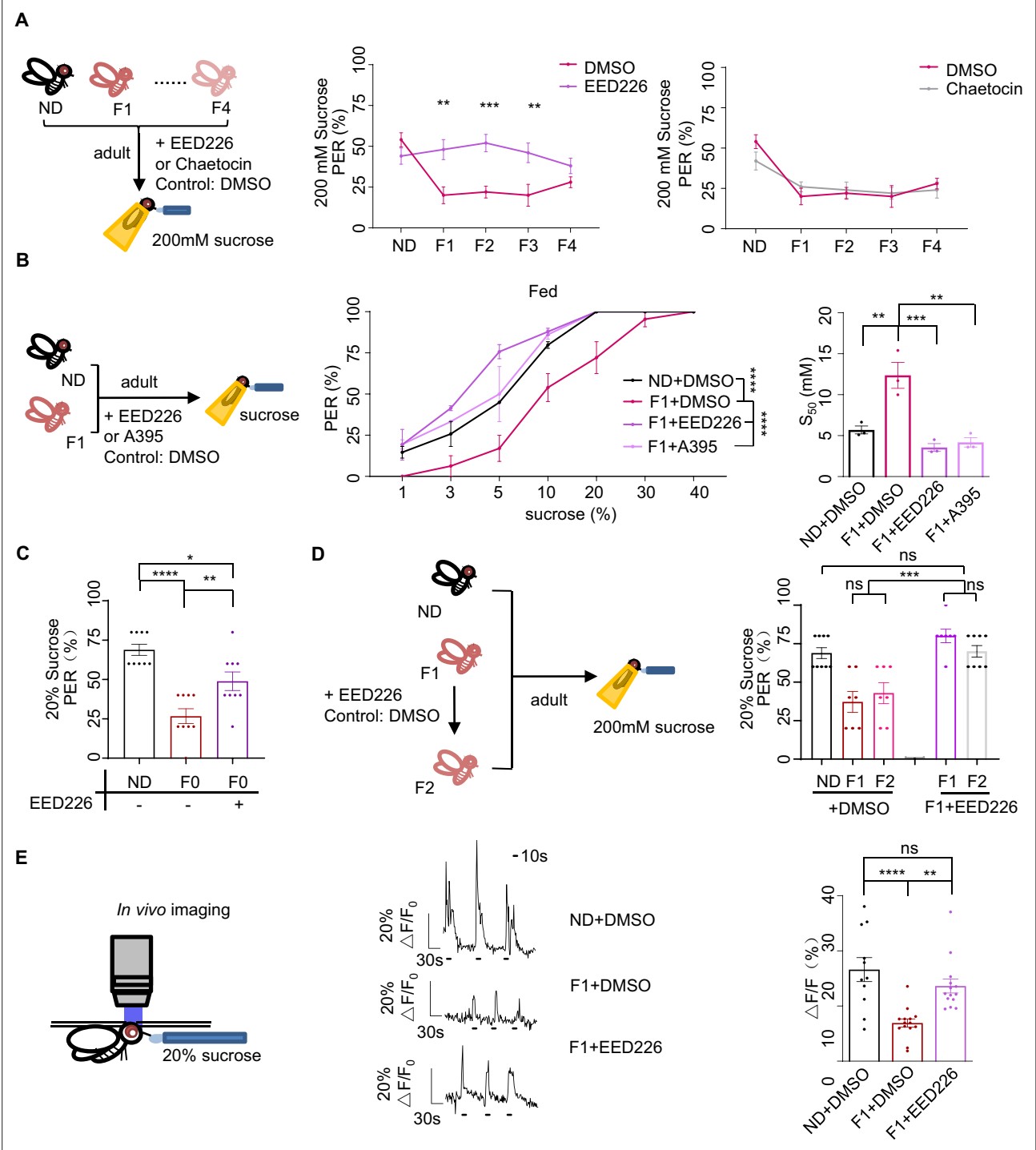

**Figure 4.** H3K27me3 modifications suppressed sweet sensitivity of Gr5a+ gustatory neurons. (**A**) Fractions of flies with or without EED226/Chaetocin feeding showing proboscis extension reflex (PER) responses to 200 mM sucrose (n=10, each containing 5 flies). Schematic diagram of experiments shown on *left*: normal diet (ND), HSD-F1 to HSD-F4 adult flies were fed with indicated chemicals for 5 days before being tested for PER response. (**B**) Fractions of flies with or without EED226/A395 feeding showing PER responses to different concentrations of sucrose (n=3–6 biological replicates, each containing 8–12 flies). The $S_{50}$ indicated the sucrose concentration that induced PER responses in 50% of the tested flies. (**C**) Fractions of flies with or without EED226 showing PER responses to 20% sucrose (n=9, each containing 5 flies). (**D**) Fractions of EED226-fed HSD-F1 flies and these offspring (HSD-F2) showing PER responses to 20% sucrose (n=7–9, each containing 5 flies). Schematic diagram of experiments shown on *left*: HSD-F1 flies were fed with indicated chemicals for 5 days, then transferred to normal medium to lay eggs to obtain HSD-F2 which mother has undergone pharmacologically treated. (**E**) The calcium signals in Gr5a+ neurons in indicated flies upon 20% sucrose. Schematic diagram of in vivo calcium

*Figure 4 continued on next page*

Figure 4 continued

imaging was shown on *left*. Representative traces of the calcium responses were shown in *middle*. Horizontal black bars represent feeding episodes. Quantification of the calcium responses was shown on *right* (n=13–16). Data were shown as means ± SEM. ns p>0.05; *p<0.05; **p<0.01; ***p<0.001; ****p<0.0001.

The online version of this article includes the following source data and figure supplement(s) for figure 4:

**Source data 1.** Raw data of the behavioral and Ca²⁺ imaging experiments shown in *Figure 4*.

**Figure supplement 1.** Inhibiting PRC2 in Gr5a⁺ gustatory neurons rescued sweet sensitivity.

**Figure supplement 1—source data 1.** Raw data of the behavioral and western blot experiments shown in *Figure 4—figure supplement 1*.

vs. ND flies (*Figure 5A*) and identified a total of 133 differentially expressed genes (DEG, |log$_{1.5}$ fold change|>1, p<0.05), including 49 downregulated genes in HSD-F1 flies (*Supplementary file 3*).

Among these genes, we noticed the homeobox gene *caudal* (*cad*), a transcription factor involved in anterior/posterior patterning, organ morphogenesis, and innate immune homeostasis (*Mlodzik and Gehring, 1987*; *Ryu et al., 2008*), and *unpaired 2* (*upd2*), a *Drosophila* leptin ortholog and a secreted factor produced by the fat body which activated JAK/STAT signaling in GABAergic neurons (*Hombría et al., 2005*; *Rajan and Perrimon, 2012*), were the only two genes located in the H3K27me3 hypermethylated genomic regions and also showed downregulated gene expression in HSD-F1 flies (*Figure 5B–E*, *Figure 5—figure supplement 1A–D*). Given that Upd2 was mainly expressed in the fat body with clear metabolic functions (*Brent and Rajan, 2020*; *Rajan and Perrimon, 2012*), we focused on ubiquitously expressed Cad for further behavioral analysis.

Notably, HSD-F1 flies exhibited a higher level of H3K27me3 modifications across the genomic region of *cad*, especially near the promoter region, whereas H3K9me3 signals exhibited no difference (*Figure 5D* and *Figure 5—figure supplement 1A*). Other signals such as H3K27ac and H3K9me2 were generally weak around the promoter region of *cad* (*Figure 5D* and *Figure 5—figure supplement 1A*). We also conducted quantitative RT-PCR analysis and confirmed that Cad expression was downregulated in both HSD-F1 and HSD-F2 flies but not in HSD-F0 flies (*Figure 5F*). Meanwhile, knockdown of E(z) and Pcl in female germline of HSD-F0 flies prevented the decline in Cad expression in HSD-F1 flies (*Figure 5G*). These data indicate that ancestral HSD exposure elevates H3K27me3 levels in the promoter region of *cad* gene, resulting in a reduction in its mRNA expression in the head tissue of offspring.

We then asked whether Cad played a direct role in regulating sweet sensitivity in Gr5a⁺ gustatory neurons. Compared to the transgenic controls, knocking down Cad expression in Gr5a⁺ neurons led to lower PER responses to sucrose (*Figure 5H*). These results were consistent with a recent study and confirmed that Cad played an important role in regulating sweet sensitivity in Gr5a⁺ neurons (*Vaziri et al., 2020*). Furthermore, these results suggest that the reduction in Cad expression likely contributes to the deficits of sweet sensitivity and feeding behavior seen in the offspring of HSD-exposed ancestors.

It was unlikely that Cad was the only gene that mediated the transgenerational behavioral effect of HSD exposure. Other potential candidate genes might be missed out in the above analysis if they did not exhibit statistically different H3K27me3 modification levels or gene expression levels (*Figure 5B*). Besides Cad, we noticed that several transcription factors, Ptx1, GATAe, nub, which were known to regulate sweet sensitivity (*Vaziri et al., 2020*), also exhibited elevated H3K27me3 modifications around their promoter regions. However, these genes didn't show downregulated gene expression in HSD-F1 flies (*Figure 5—figure supplement 2A and B*). It is thus possible that multiple transcriptional regulators are epigenetically modified by ancestral HSD exposure, which in turn exerts transgenerational effects on the sweet sensitivity and feeding behavior in offspring.

## Discussion

Transgenerational behavioral change is present in many animal species. In *C. elegans*, exposure to pathogenic threats induces avoidance memories which can transmit for four generations via sRNA signaling (*Moore et al., 2021*). In fruit flies, exposure to predatory wasps leads to the inheritance of ethanol preference for five generations via maternal NPF repression (*Bozler et al., 2019*). Such a 'behavior memory' may be evolutionarily beneficial in a sense to pre-adapt offspring for changing

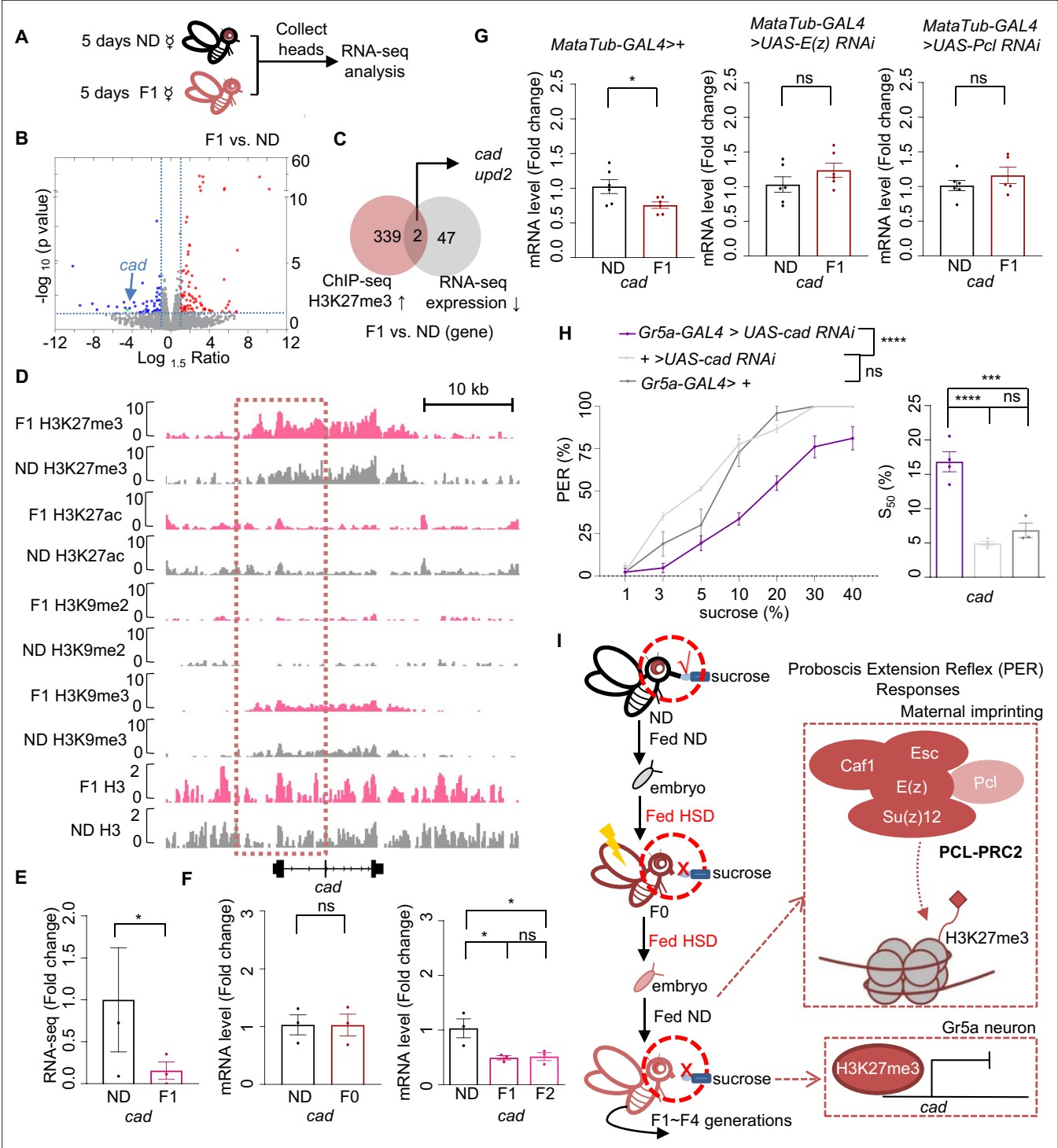

**Figure 5.** Cad mediated the inhibitory effect of H3K27me3 modifications on sweet sensitivity and feeding behavior. (**A**) Preparation of RNA samples for RNA-seq. (**B**) Volcano plot showed differentially expressed genes between normal diet (ND) and HSD-F1 flies (blue: downregulated genes in HSD-F1 flies; red: upregulated genes in HSD-F1 flies; green: *cad*). The horizontal line indicated the significance threshold (p=0.05) and the vertical lines indicated the 1.5-fold change threshold. (**C**) Venn diagram of downregulated genes and H3K27me3-target genes between ND and HSD-F1 flies (gray: downregulated genes in the heads of HSD-F1 flies via RNA-seq analysis, fold change >1.5; red: H3K27me3-target genes in HSD-F1 embryos via chromatin-immunoprecipitation followed by sequencing [ChIP-seq] analysis, $\log_{10}$ likelihood ratio >1). (**D**) Genome browser view of H3K27me3, H3K27ac, H3K9me2, H3K9me3, and H3 density around the *cad* gene in embryos of ND and HSD-F1 flies. (**E–G**) mRNA expression levels of *cad* in indicated flies. Fly heads were collected and subjected to RNA-seq (**E**) (n=3 biological replicates, each containing 15 fly heads) or quantitative RT-PCR (**F** and **G**) (n=3–6 biological replicates, each containing 15 fly heads). (**H**) Fractions of flies of the indicated genotypes showing PER responses to different

*Figure 5 continued on next page*

*Figure 5 continued*

concentrations of sucrose (n=3–6 biological replicates, each containing 8–12 flies). The $S_{50}$ indicated the sucrose concentration that induced PER responses in 50% of the tested flies. (I) A working model: Upon ancestral HSD exposure, PCL-PRC2 complex was activated and maintained high levels of H3K27me3 imprinting in maternal germline and the offspring. H3K27me3 targeted *cad* promoter regions and suppressed its expression, resulting in an inhibitory effect in Gr5a[+] sweet-sensing gustatory neurons. Consequently, both sweet sensitivity and feeding behavior in the offspring were suppressed. Data are shown as means ± SEM. ns p>0.05; *p<0.05; **p<0.01; ***p<0.001; ****p<0.0001.

The online version of this article includes the following source data and figure supplement(s) for figure 5:

**Source data 1.** Raw data of the RNA-seq, qPCR, and behavioral experiments shown in *Figure 5*.

**Figure supplement 1.** H3K27me3 modifications and mRNA expression of *cad* and *upd2* genes upon ancestral high-sugar diet (HSD) exposure.

**Figure supplement 1—source data 1.** Raw data of the RNA-seq and qPCR experiments shown in *Figure 5—figure supplement 1*.

**Figure supplement 2.** H3K27me3 modifications and mRNA expression of *Ptx1, GATAe, and nub* upon ancestral high-sugar diet (HSD) exposure.

**Figure supplement 2—source data 1.** Raw data of the RNA-seq experiments shown in *Figure 5—figure supplement 2*.

environmental conditions. Nevertheless, it may also lead to devastating effects in human health. Individuals who were exposed to the Dutch Famine in early gestation exhibited deficits in metabolism, cardiovascular health, and mental health (*Schulz, 2010*). Similarly, mice exposed to traumatic experiences and drugs induce depressive-like or autism-like behaviors in their progeny (*Chan et al., 2020*; *Choi et al., 2016*; *Gapp et al., 2014*).

Globally, HSD has become a routine of modern lifestyle, which is linked to various human diseases, including obesity, type 2 diabetes, and neurobiological diseases (*Malik and Hu, 2022*; *Malik et al., 2010*). HSD also induces many behavioral disorders in animal models such as feeding abnormalities and addiction-like behaviors (*Avena et al., 2009*; *May et al., 2019*). This present study uncovers that HSD not only affects flies' overall physiology and metabolism within the same generation, but also affects their sweet sensitivity and feeding behavior in a manner spanning multiple generations of offspring. If similar observations hold true in human society, HSD exposure may lead to an additional layer of health risk that needs to be recognized and addressed.

Mechanistically, our data indicate that elevated H3K27me3 modifications upon ancestral HSD exposure are the key epigenetic factors underlying the transgenerational regulations of sweet sensitivity and feeding behavior (*Figure 5I*). HSD exposure enhances genome-wide H3K27me3 but not H3K27ac or H3K9me2/3 modifications in early embryos. E(z) and Pcl, the key components of PCL-PRC2 complex, play a crucial role in catalyzing trimethylation of the repressive chromatin marker histone H3 lysine 27, thus maintaining this imprinting during fly development. Perturbation of PCL-PRC2 complex, via both genetic and pharmacological approaches, blocks the transmission of such repressive histone imprinting to offspring and eliminates the transgenerational modulation of sweet sensitivity and feeding behavior. Such epigenetic modulations are transmitted via the female germline, and exert a long-lasting effect on the expression of Cad (and possibly other transcriptional regulators) in the offspring. Cad, a transcription factor belonging to the Hox family, regulates the sensitivity of sweet-sensing gustatory neurons and plays a role in modulating PER responses to sucrose.

Feeding behavior is tightly regulated by various factors, such as internal nutritional needs, overall physiological status, and environmental cues. However, most of these regulations are quite dynamic in nature and do not last for long. For example, in fruit flies energy shortage can trigger foraging and feeding behavior in the timescale of hours, which can be rapidly suppressed upon acquisition of desirable food sources (*Basiri and Stuber, 2016*; *Qi et al., 2021*). These regulations are often mediated by rapid-acting molecular and cellular mechanisms such as ion channels, hormones, and neuropeptides (*Pool and Scott, 2014*). HSD can also modulate feeding behavior in a dynamic manner (*Sung et al., 2022*). Moreover, this mechanism is reversible within the same generation. For example, the number of PLCβ2[+] taste bud cells in the fungiform papilla decreased within 4 weeks of HSD exposure in mice and was completely restored within 4 weeks upon the removal of HSD (*Sung et al., 2022*). However, our present study reveals a highly persistent and heritable mechanism of transgenerational modulation of sweet taste perception and feeding behavior that can transmit through multiple generations without re-exposure to the original HSD treatment. These findings suggest that feeding behavior can be modulated by different regulatory mechanisms with distinct timescales in response to various types of environmental and internal state changes. The relationship between these different mechanisms will be of great interest to further study. For example, it remains unclear whether and how hormonal

and neurotransmitter changes that occur upon HSD exposure in flies within the same generation contribute to the reprogramming of H3K27me3 that can last for multiple generations.

Based on our present study, several important questions remain to be answered. The first question is how HSD evaluates H3K27me3 modifications. Previous work reported that some dietary bioactive compounds could regulate histone modifying enzymes (*Vahid et al., 2015*). Thus, HSD may directly regulate the activity of histone methyltransferases or histone demethylases. Alternatively, HSD may affect H3K27me3 modifications via certain nutrient-sensing mechanisms such as *O*-GlyNAc transferase, which is known to facilitate H3K27me3 formation with PRC2. The second question lies in how elevated H3K27me3 modifications in specific gene loci are retained during gamete formation and embryo development, and how such epigenetic imprinting is removed after four to five generations on ND. Previous work reported that long-term memory of H3K27me3 depends on efficient copying of this mark after each DNA replication cycle in a PRC2-dependent manner (*Coleman and Struhl, 2017*). It is therefore possible that PRC2-mediated maintenance of H3K27me3 imprinting can be regulated by dietary exposure. The third question is how H3K27me3 imprinting affects specific neurons in the adult via Cad signaling. We showed that upregulated H3K27me3 modifications decreased Cad expression. Future studies are needed to examine how Cad signaling impacts the development and function of Gr5a$^+$ gustatory neurons, and whether other transcription factors may also be involved. Previous work has reported that Cad regulates a network containing 119 candidate genes that were implicated in sensory perception of chemical stimulus, neuropeptide signaling pathways, signal transduction, and transcription factor activity, which could impact the sweet gustatory neurons and play a role in modulating PER responses to sucrose (*Vaziri et al., 2020*).

In this present study we focus on the function of histone modifications on transgenerational behavioral inheritance. It will also be of interest to explore other mechanisms that may also participate in the transgenerational inheritance of sweet perception. Especially, as the Piwi protein negatively regulates H3K27 trimethylation (*Peng et al., 2016*), future studies are needed to understand whether piRNAs, which have been reported to direct transient heterochromatin formation and stabilize maternal mRNAs during embryogenesis (*Dufourt et al., 2017*; *Fabry et al., 2021*; *Wang and Lin, 2021*), are involved in the transgenerational inheritance we identified. Given HSD exposure imposes a systemic influence on flies' physiology and metabolism, other non-epigenetic factors, such as maternal nutritional conditions and their potential influences on the development and maturation of oocytes, may also play a role in the transgenerational inheritance of sweet perception upon ancestral HSD exposure. Furthermore, the identification of HSD exposure-induced transgenerational changes in sweet sensitivity and feeding raises the question of whether similar phenomena and mechanism can be extended to other behaviors. Our findings highlight a novel and pivotal role of epigenetic modifications in preparing animals for the dynamic environment, which opens a new avenue of research to further uncover the interactions among prior experience, epigenetics, and behavioral modulations across generations.

# Materials and methods

## Key resources table

| Reagent type (species) or resource | Designation | Source or reference | Identifiers | Additional information |
|---|---|---|---|---|
| Genetic reagent (*D. melanogaster*) | *nosNGT-GAL4* | Bloomington Drosophila Stock Center | Cat: #31777 RRID:BDSC_31777 | FlyBase symbol: P{Gal4-nos.NGT}40 |
| Genetic reagent (*D. melanogaster*) | *Maternal-tubulin-Gal4* | Bloomington Drosophila Stock Center | Cat: #2318 PMID:23105012 | FlyBase symbol: P{matα4-GAL-VP16} |
| Genetic reagent (*D. melanogaster*) | *UAS-GCaMP6m* | Bloomington Drosophila Stock Center | Cat: #42748 RRID:BDSC_42748 | FlyBase symbol: w[1118]; P{y[+t7.7] w[+mC]=20XUAS-IVS-GCaMP6m} attP40 |
| Genetic reagent (*D. melanogaster*) | *Gr5a-GAL4* | Bloomington Drosophila Stock Center | Cat: #57592; RRID:BDSC_57592 | FlyBase symbol: w[*]; P{w[+mC]=Gr5a-GAL4.8.5}6; Dr[1]/TM3, Sb |
| Genetic reagent (*D. melanogaster*) | *UAS-E(z) RNAi* | Tsinghua Fly Center | Cat: #2831 | |

*Continued on next page*

*Continued*

| Reagent type (species) or resource | Designation | Source or reference | Identifiers | Additional information |
|---|---|---|---|---|
| Genetic reagent (*D. melanogaster*) | *UAS-Su(var)3–9 RNAi* | Tsinghua Fly Center | Cat: #3558 | |
| Genetic reagent (*D. melanogaster*) | *UAS-Rpd3 RNAi* | Tsinghua Fly Center | Cat: #0695 | |
| Genetic reagent (*D. melanogaster*) | *UAS-cad RNAi* | Tsinghua Fly Center | Cat: #03,877.N | |
| Antibody | Anti-Histone H3 (di methyl K9) mouse polyclonal | Abcam | Cat: #ab1220 RRID:AB_449854 | ChIP-seq 10 µg |
| Antibody | Anti-acetyl histone-h3-k27 rabbit polyclonal | Abcam | Cat: #ab4729 RRID:AB_2118291 | ChIP-seq 10 µg |
| Antibody | Anti-Histone H3 (tri methyl K27) mouse polyclonal | Abcam | Cat: #ab6002 RRID:AB_305237 | ChIP-seq 10 µg |
| Antibody | Anti-Histone H3 (tri methyl K9) rabbit polyclonal | Abcam | Cat: #ab8898 RRID:AB_306848 | ChIP-seq 10 µg |
| Antibody | Anti-Histone H3 antibody rabbit polyclonal | Abcam | Cat: #ab1791 RRID:AB_302613 | ChIP-seq 10 µg WB 1:1000 |
| Antibody | Anti-Histone H3 (tri methyl K27) rabbit polyclonal | Jingjie PTM BioLab | Cat: #PTM-647RM | WB 1:1000 IF 1:200 |
| Antibody | Anti-rabbit IgG, HRP-linked Antibody goat polyclonal | Jackson ImmunoResearch Laboratories Inc | Cat: #111-035-003 RRID: AB_2313567 | WB 1:1000 |
| Antibody | Anti-Histone H3 (tri methyl K27) mouse polyclonal | Jingjie PTM BioLab | Cat: #PTM-5002 | WB 1:1000 IF 1:200 |
| Antibody | Goat anti-mouse IgG (H+L) Cross-Adsorbed Secondary Antibody, Alexa Fluor 488 goat polyclonal | Invitrogen | Cat: #a11001 RRID: AB_2534069 | IF 1:1000 |
| Antibody | Goat anti-Rabbit IgG (H+L) Highly Cross-Adsorbed Secondary Antibody, Alexa Fluor 633 goat polyclonal | Invitrogen | Cat: #a21071 RRID: AB_2535732 | IF 1:1000 |
| Chemical compound | Fluoroshield with DAPI | Sigma-Aldrich | Cat: #F6057 | 10 µL |
| Chemical compound | EED226 | Selleck | Cat: #S8496 | 5 µM |
| Chemical compound | Chaetocin | Selleck | Cat: #S8068 | 5 µM |
| Chemical compound | A-395 | Sigma-Aldrich | Cat: #SML1923 | 10 µM |
| Commercial assay or kit | Graduated glass capillary | VWR | Cat: #53432-604 | |
| Commercial assay or kit | Triglyceride Quantification Colorimetric/ Fluorometric Kit | Nanjing Jiancheng Bioengineering Institute | Cat: #A110-1-1 | |
| Commercial assay or kit | Liver/Muscle glycogen assay kit | Nanjing Jiancheng Bioengineering Institute | Cat: #A043-1-1 | |
| Commercial assay or kit | Trehalose quantification kit | Nanjing Jiancheng Bioengineering Institute | Cat: #A149-1-1 | |
| Commercial assay or kit | MOPS Running Buffer Powder MOPS | GenScript Biotech Corporation | Cat: #M00138 | for Western blotting |
| Commercial assay or kit | ExpressPlus PAGE Gel,10×8, 4–20%, 15 wells | GenScript Biotech Corporation | Cat: #M42015c | for Western blotting |
| Commercial assay or kit | SDS-PAGE Sample Loading Buffer, 5× | Biosharp | Cat: #BL502A | for Western blotting |
| Commercial assay or kit | Affinity ECL kit(picogram) | Affinity | Cat: #KF8001 | for Western blotting |
| Commercial assay or kit | Goat Serum | Beyotime | Cat: #C0265 | for Western blotting |
| Chemical compound | Hexanoic acid | macklin | Cat: #H810882 | |
| Chemical compound | Sucrose | Sigma-Aldrich | Cat: #S0389 | |
| Chemical compound | NaCl | Sigma-Aldrich | Cat: #S3014 | |
| Chemical compound | $CaCl_2$ | Sigma-Aldrich | Cat: #C5670 | |

*Continued on next page*

*Continued*

| Reagent type (species) or resource | Designation | Source or reference | Identifiers | Additional information |
|---|---|---|---|---|
| Chemical compound | MgCl₂ | Sigma-Aldrich | Cat: #M4880 | |
| Chemical compound | KCl | Sigma-Aldrich | Cat: #P9541 | |
| Chemical compound | HEPES | Sigma-Aldrich | Cat: #54457 | |
| Chemical compound | NaHCO₃ | Sigma-Aldrich | Cat: #S5761 | |
| Chemical compound | NaH₂PO₄ | Sigma-Aldrich | Cat: #S3139 | |
| Software, algorithm | Fiji/ImageJ | NIH | | https://imagej.nih.gov/ij/download.html |
| Software, algorithm | Rstudio | | | https://www.rstudio.com/ |
| Software, algorithm | GraphPad Prism 9 | GraphPad Software | | https://www.graphpad.com/scientificsoftware/prism/ |
| Software, algorithm | R version 3.6 and 4.0 | Camp Pontanezen – The R Foundation for Statistical Computing | | https://www.r-project.org/ |
| Software, algorithm | Bowtie2 version 2.3.5.1 | | | https://bowtie-bio.sourceforge.net/bowtie2/index.shtml |
| Software, algorithm | samtools version 1.10. | | | http://www.htslib.org/ |
| Software, algorithm | deeptools version 3.3.1 | | | https://deeptools.readthedocs.io/en/develop/ |
| Software, algorithm | IGV version 2.4.13 | | | https://software.broadinstitute.org/software/igv/download |

## Flies

Flies were kept in vials containing a standard medium made of yeast, corn, and agar at 25°C, 60% relative humidity, and on a 12 hr light-12-hr dark cycle. Virgin female flies were collected shortly after eclosion and kept in groups (25 flies per vial) on standard fly medium (ND, with 10% sucrose) or HSD (ND plus an additional 10% sucrose) for 4–6 days before experiments.

Fly strains used in the manuscript: *nosNGT-GAL4:* (#31777), *Maternal-tubulin-Gal4* (#2318), *Gr5a-GAL4* (#57592), and *UAS-GCaMP6m* (#42748) were obtained from the Bloomington Drosophila Stock Center at Indiana University; *UAS-E(z) RNAi* (#2831), *UAS-Pcl RNAi* (#1185), *UAS-Su(var)3–9 RNAi* (#3558), *UAS-Rpd3 RNAi* (#0695), and *UAS-cad RNAi* (#03,877.N) were from the Tsinghua Fly Center.

## Chemicals and antibodies

Sucrose (#S0389), NaCl (#S3014), CaCl₂ (#C5670), MgCl₂ (#M4880), KCl (#P9541), NaH₂PO₄ (#S3139), NaHCO₃ (#S5761), and HEPES (#54457) were from Sigma-Aldrich. Hexanoic acid (Macklin, #H810882) for PER assay was purchased from Macklin. EED226 (5 µM, Selleck, #S8496), Chaetocin (5 µM, Selleck #S8068), and A-395 (10 µM, Sigma-Aldrich, #SML1923) were added to the standard medium. Flies were kept on these foods for 5 days before the assay (change fresh medium every 2 days).

The following antibodies were used: The antibodies against Histone H3 (di methyl K9) (#ab1220), acetyl histone-h3-k27 (#ab4729), Histone H3 (trimethyl K27) (#ab6002, #PTM-5002, #PTM-647RM), Histone H3 (trimethyl K9) (#ab8898), and Histone H3 antibody (#ab1791) were purchased from Abcam and Jingjie PTM BioLab. The secondary antibodies against rabbit (Alexa Fluor 633, #a21071, HRP-linked Antibody, #111-035-003) and mouse (Alexa Fluor 488, #a11001) were purchased from Jackson ImmunoResearch and Invitrogen. Fluoroshield with DAPI (#F6057) was purchased from Sigma-Aldrich.

## Triglyceride, glycogen, and trehalose measurement

For triglyceride and glycogen, a single fly was anesthetized and transferred to tube with the corresponding extract. The samples were measured according to the manufacturer's instructions. Glycogen was measured with Liver/Muscle glycogen assay kit (#A043-1-1, Nanjing Jiancheng Bioengineering Institute, China). Triglyceride was measured with Triglyceride Quantification Colorimetric/Fluorometric Kit (#A110-1-1, Nanjing Jiancheng Bioengineering Institute).

The hemolymph trehalose was quantified by trehalose quantification kit (#A149-1-1, Nanjing Jiancheng Bioengineering Institute). Briefly, 40 flies were anesthetized and then pierced in the thorax with dissecting forceps. The pierced flies were then transferred to perforated tubes and centrifuged for 5 min at 3000×$g$ at 4°C to collect the hemolymph. Afterward, 0.6 μL hemolymph was quickly removed into a 200 μL tube with trehalose extract and vortexed for 2–3 min. After 45 min standing, the sample was centrifuged at 8000×$g$ for 10 min. Then 175 μL supernatant was added to 700 μL reaction solution and boiled for 5 min. After cooling, 250 μL sample was removed to a 96-well plate for the light absorption value measurement at 620 nm.

## The CAFE assay

As described previously (*Ja et al., 2007*), 25 indicated virgin flies were collected upon eclosion and aged for 4 days. To construct the CAFE setup, one hole was bored into the lid of a *Drosophila* bottle and a 5 μL glass capillaries (VWR, #53432-604) filled with 20% sucrose was inserted. The bottle also contained 2% agar medium to ensure satiation with water. 10 female virgin flies were inserted into each bottle by mouth aspiration and adapted for 24 hr, then the capillaries were changed to new ones containing same concentration sucrose, and the level of capillary was marked. After 24 hr at 25°C, 60% relative humidity, the level of capillary was marked again and the distance between these marks was converted into a volume consumed per fly. In addition, three blank bottles without flies were set up in the same way, and the mean volume change from these capillaries was subtracted from the capillaries with flies, to control for the effect of evaporation.

## The MAFE assay

As described previously (*Qi et al., 2015*), individual flies were transferred and immobilized in a 200 μL pipette tip, and then sated with sterile water before being presented with 400 mM sucrose filled in a graduated glass capillary (VWR, #53432-604). The food stimulation was repeated until the flies became unresponsive to a series of 10 food stimuli, and the total food consumption was calculated based on the volume change during feeding process.

## PER assay

PER assay was performed as described (*Qi et al., 2015*). Briefly, individual flies were gently aspirated and immobilized in a 200 μL pipette tip as in the MAFE assay. Flies were first sated with water and then subjected to different sugar or fatty acid solutions with each solution tested twice. Flies showing PER responses to at least one of the two trials were considered positive to that sugar or fatty acid concentration. Fatty acids were dissolved in ethanol and tested at a concentration of 1% (*Ahn et al., 2017*; *Brown et al., 2021*). All PER experiments had n>3 replicates with 8–12 flies per replicate or n>6 replicates with 5 flies per replicate unless otherwise stated.

The $S_{50}$ indicates the sucrose concentration that induces 50% PER, which was estimated using the basic linear or nonlinear regression model based on a previous method (*Vaziri et al., 2020*). All $S_{50}$ estimations were performed in R package using the 'basicTrendline' function.

## Quantitative RT-PCR

Total RNA was extracted from the head tissues of flies. RNA was reverse-transcribed with the All-in-One cDNA Synthesis Supermix (TransScript). Quantitative RT-PCR was conducted on Bio-Rad CFX96 using the SYBR green PCR master mix (TaKaRa, Japan) with the primers listed in quantitative RT-PCR primers table. Relative mRNA levels were calculated relative to *rp49* expression by the comparative Ct method.

### Quantitative RT-PCR primers table

| Gene | Primers |
| --- | --- |
| *rp49* | CGCACCAAGCACTTCATCC |
| | ACGCACTCTGTTGTCGATACC |

*Continued on next page*

*Continued*

| Gene | Primers |
|------|---------|
| *Pcl* | CGGGAATTTGTCGCCAGTTG |
| | TCGTTGACCCGATGCTTCTC |
| *E(z)* | ATGCTGACCAAGACCTGTCG |
| | GGAGGTGTGAAGTCCTGTCG |
| *Su(z)12* | GCCAGCAACCAGTAACAACG |
| | CCTTGGTCCTCTCCGATGTG |
| *Esc* | GCTGGAAACCGGGACAACTA |
| | AGCCGAATCTCACGAACCAG |
| *Caf-1* | AGCCTCGAAATGGTGGATCG |
| | GTCGAAGGATTCCGCTGCTA |
| *Dilp2* | GCCTTTGTCCTTCATCTCG |
| | CCATACTCAGCACCTCGTTG |
| *Dilp5* | TTTAGGCAAATGAAATACGGC |
| | AACGCAGCCGATACTCACA |
| *cad* | CGACTCAAGTTTGCCTTATTTATTA |
| | TTTAGGCAAATGAAATACGGC |
| *upd2* | TTCCTGCCGAACATGACGAG |
| | GGTCCGCTTCACTCTGTCTC |
| *Utx* | GCTCAGTCAAGCACCATTGC |
| | AGCATCTGCGCTGTTTGTTG |
| *Cbp* | ACTTGGGAAGAGCAGTTCGG |
| | CGATGCGTTTGGCCATCTTC |

## Calcium imaging

For in vivo imaging (*Yang et al., 2018*), flies were anesthetized on ice and glued onto transparent tape. Then, a hole (~1–2 mm) on the tape was incised to expose the dorsal part of the fly head. The cuticle part around the Gr5a[+] neuron regions of the fly brain was gently removed with forceps and the brain was bathed in the adult hemolymph-like solution (108 mM NaCl, 8.2 mM $MgCl_2$, 4 mM $NaHCO_3$, 1 mM $NaH_2PO_4$, 2 mM $CaCl_2$, 5 mM KCl, 5 mM HEPES, 80 mM sucrose, pH 7.3). A micro manipulator delivered liquid food to the proboscis of the fly at the indicated time and the actual feeding bouts were imaged by a digital camera installed under the imaging stage at 0.5 frame/s.

More specifically, at each feeding bout, the flies extended their proboscis to reach the surface of the liquid food and started food ingestion. By adding a blue dye in the liquid food, the actual flow of the dyed food through flies' pharynx could also be seen.

The calcium signals of Gr5a[+] neurons were recorded by a Nikon C2 confocal microscope, with a water immersion objective lens (40×/0.80 w DIC N2) at 0.2 frame/s.

Image analyses were performed in ImageJ and plotted in Excel (Microsoft). The ratio changes were calculated using the following formula: $\Delta F/F = [F - F_0]/F_0$, where F is the mean florescence of cell body, $F_0$ is the average base line (~60 s interval before stimulation).

## RNA-seq and analysis

Total RNA from fly heads was extracted from 5-day-old female flies using the Trizol reagent (Invitrogen, USA). mRNA was purified from total RNA using oligo(dT)-attached magnetic beads, followed by library preparation (the quality of libraries was checked by Bioanalyzer 2100 [Agilent]) and sequencing

(BGISEQ 500 platform) with paired-end 150 bps. Sequence data were subsequently mapped to *Drosophila* genome and uniquely mapped reads were collected for further analysis. Gene expression was calculated by the FPKM (fragments per kilobase of exon per million fragments mapped). The genes with a p-value less than 0.05 and $|\log_{1.5}$fold change| more than 1 were considered as differentially expressed genes. The RNA-seq data were deposited in GEO database under the accession codes (GSE216075 and GSE215756).

## Embryo sorting

Flies were maintained in large population cages in an incubator set at standard conditions (25°C). Cycle 10–12 embryos for ChIP-seq were collected for 30 min, and then allowed to develop for 50 additional minutes before being harvested (*Foe and Alberts, 1983*). Embryos for immunofluorescence staining were collected for 2 hr, and then allowed to develop for 30 additional min before being harvested (*Huang et al., 2017*). Harvested embryos were dechorionated for 2 min in 50% sodium hypochloride and hand sorted (within 30 min) in a small dish using an inverted microscope to remove embryos younger or older than the targeted age range based on morphology of the embryos as previously described (*Li et al., 2014*).

All embryos were preserved in the embryo stock buffer A1 (60 mM KCl, 15 mM NaCl, 4 mM MgCl$_2$, 15 mM HEPES, 0.5% Triton X-100, 0.5 mM DTT, and 10 mM sodium butyrate) at –80°C or frozen by liquid nitrogen.

## Immunofluorescence staining

For immunofluorescence staining (*Iovino et al., 2013*; *Tang et al., 2020*), germarium was dissected from 3-day-old virgin flies in PBS and fixed in 4% PFA at room temperature for 1 hr. Harvested embryos were fixed in heptane. The fixed germarium and embryos were rinsed in PBST (PBS with 0.3% Triton X-100) three times and blocked in PBST with 5% normal goat serum for 1 hr at room temperature. The tissues were then incubated with primary antibody (anti-H3K27me3 #PTM-647RM 1:200, #PTM-5002 1:200) diluted in PBST with 5% normal goat serum overnight at 4°C. After being rinsed three times in PBST, the tissues were incubated with appropriate fluorescently labeled secondary antibodies (goat anti mouse 488 #a11001 1:1000, goat anti rabbit 633 #a21071 1:1000) in dark for 1 hr. They were then stained with DAPI (Fluoroshield with DAPI, Sigma, #F6057). Images were collected on the ZEISS LSM800 confocal system.

## Western blotting

For western blot (*Huang et al., 2020*), fly tissues and embryos were collected and homogenized in PBS with protease inhibitors. Samples were denatured, separated by SDS-PAGE, and transferred to a polyvinylidene difluoride membrane. After being blocked in TBST containing 5% milk, the membrane was incubated with the specific primary antibody (anti-H3 #ab1791 1:1000, anti-H3K27me3 #PTM-647RM, 1:1000) followed by HRP-conjugated goat anti-rabbit (#111-035-003, 1:5000). The specific bands were detected by an ECL western blotting detection system (Bio-Rad, USA). Qualification was performed using Image J software.

## ChIP-seq and analysis

Formaldehyde was added to the embryos for cross-linking for 10 min, then glycine was added to quench the formaldehyde, followed by centrifugation and removal of the supernatant. The pellet was washed twice and then lysed with the Lysis Buffer (140 mM NaCl, 15 mM HEPES, 1 mM EDTA, 0.5 mM EGTA, 1% Triton X-100, 0.1% sodium deoxycholate). The lysate is sonicated using Qsonica (duty cycle – 10%; intensity – 5; cycles per burst – 200; time – 4 min) and centrifuged. The chromatin obtained was fragmented to sizes ranging from 100 to 300 bp. The supernatant was carefully transferred to a new tube to incubate with corresponding antibody (10 µg) at 4°C overnight. Then, 15 µL of Pierce Protein A/G beads (Thermo Fisher) were added to the mixture, followed by further incubation for 4 hr on a rotator. After washing four times with TE Buffer (0.1 mM EDTA, 10 mM Tris-HCl pH 8.0), the beads were incubated with the 250 µL Elution Buffer (10 mM EDTA, 50 mM Tris HCl pH 8.0, 1% SDS) at 65°C for 30 min. Then, 1.6 µL 25 mg/mL RNase A (Sigma) was added, and all samples were incubated at 65°C for 3 hr. After the RNase A treatment, all samples were further treated with 6 µL Proteinase K (Sigma) at 56°C for 2 hr. The resulting DNA was purified by Agencourt Ampure beads (Beckman

Coulter). One µg of DNA was used to generate sequencing library using the mRNA-Seq Sample Preparation Kit (Illumina) and sequenced on an Illumina Hiseq platform (Novagene) with paired-end 300 bps.

Raw reads were cleaned using trim galore. The reads were then aligned to the dm6 genome assembly using Bowtie2 v2.3.5.1 with default parameters. Duplicate reads were then removed using MarkDuplicates from gatk package v.4.1.4.1. Replicate samples were merged using Samtools v1.10. For ChIP-seq, bigwig tracks were generated using bamCompare from deepTools 3.3.1 (parameters: --skipNAs --scaleFactorsMethod CPM --operation log2 --extendReads 200). Negative values were set to zero. Peak calling was performed using Macs2 v2.2.6 callpeak with default parameters. ChIP-seq profiles were created by computeMatrix and plotProfile in deepTools 3.3.1. IGV v.2.4.13 was used to visualize the bigwig tracks. The difference peaks of ChIP-seq data were found by using MACS2 with the options 'bdgdiff'. The parameters were '--t1 --c1 --t2 --c2 --d1 --d2 --o-prefix' and others were default.

## Statistical analysis

Data are represented as means ± SEM. Statistical tests were performed using D'Agostino-Pearson omnibus test for normal distribution, t test or Mann-Whitney U-test for two groups comparisons, one-way ANOVA, two-way ANOVA with a Bonferroni correction (Bonferroni post hoc test), or Kruskal-Wallis H test for comparisons among three or more groups and comparisons with more than one variant. All statistical analysis was performed using GraphPad Prism 9.0.

## Acknowledgements

We thank all Wang Lab members for helpful discussions and technical assistance. We thank Dr. Wei Xie from Tsinghua University and Neuroscience Pioneer Club for helpful discussions throughout the course of this study. Ye Wu and Tingting Song provided scientific and administrative support in the laboratory. We thank the Bloomington Drosophila Stock Center at Indiana University and the Tsinghua Fly Center for fly stocks and reagents. This study was funded by National Key R&D Program of China (2019YFA0802400 and 2019YFA0801900), the National Natural Science Foundation of China (32071006), and the startup funds from Shenzhen Bay Laboratory.

## Additional information

### Funding

| Funder | Grant reference number | Author |
| --- | --- | --- |
| National Key Research and Development Program of China | 2019YFA0802400 | Liming Wang |
| National Key Research and Development Program of China | 2019YFA0801900 | Liming Wang |
| National Natural Science Foundation of China | 32071006 | Liming Wang |

The funders had no role in study design, data collection and interpretation, or the decision to submit the work for publication.

### Author contributions

Jie Yang, Resources, Data curation, Formal analysis, Validation, Investigation, Methodology, Writing - original draft; Ruijun Tang, Resources, Formal analysis, Investigation, Methodology; Shiye Chen, Yinan Chen, Investigation; Kai Yuan, Resources, Supervision, Methodology; Rui Huang, Resources, Formal analysis, Supervision, Funding acquisition, Investigation, Methodology, Writing - original draft, Project administration, Writing - review and editing; Liming Wang, Conceptualization, Resources, Supervision, Funding acquisition, Investigation, Writing - original draft, Project administration, Writing - review and editing

## Author ORCIDs

Jie Yang http://orcid.org/0000-0002-0833-9661
Yinan Chen http://orcid.org/0000-0002-5543-3976
Kai Yuan http://orcid.org/0000-0001-7002-5703
Rui Huang http://orcid.org/0000-0003-4656-1682
Liming Wang http://orcid.org/0000-0002-7256-8776

## Decision letter and Author response

Decision letter https://doi.org/10.7554/eLife.85365.sa1
Author response https://doi.org/10.7554/eLife.85365.sa2

## Additional files

### Supplementary files

• Supplementary file 1. H3K27me3 upregulation and downregulation regions in the early embryo upon ancestral high-sugar diet (HSD) exposure ($\log_{10}$ likelihood ratio >3). The cycle 10–12 embryos of HSD-F1 vs. normal diet (ND) flies were collected and subjected to chromatin-immunoprecipitation followed by sequencing (ChIP-seq) analysis. The difference peaks of ChIP-seq data were found by using model-based analysis of ChIP-seq 2 (MACS2) with the options 'bdgdiff'. The cutoff is $\log_{10}$ likelihood ratio >3 (n=2 biological replicates, each containing 1000 embryos).

• Supplementary file 2. H3K27me3 upregulation and downregulation regions in the early embryo upon ancestral high-sugar diet (HSD) exposure ($\log_{10}$ likelihood ratio >1). The original data of this experiment were from the same batch as *Supplementary file 1*, but the differential peaks were reanalyzed. The cutoff is $\log_{10}$ likelihood ratio >1 (n=2 biological replicates, each containing 1000 embryos).

• Supplementary file 3. Differentially expressed genes in the adult head tissues upon ancestral high-sugar diet (HSD) exposure. The head tissues of HSD-F1 vs. normal diet (ND) flies were collected and subjected to RNA-seq analysis. The difference gene of RNA-seq data were found by using DEseq2. The cutoff is p-value less than 0.05 and $|\log_{1.5}\text{fold change}|$ more than 1 (n=3 biological replicates, each containing 15 fly heads).

• MDAR checklist

### Data availability

Sequencing data have been deposited in GEO under accession codes GSE216075 and GSE215756. All data generated or analysed during this study are included in the manuscript and supporting file; Source Data files have been provided for all figures and figure supplements.

The following datasets were generated:

| Author(s) | Year | Dataset title | Dataset URL | Database and Identifier |
|-----------|------|---------------|-------------|-------------------------|
| Yang J | 2022 | Histone methylation mediates transgenerational modulations of sweet perception by high sugar diet [ChIP-Seq] | https://www.ncbi.nlm.nih.gov/geo/query/acc.cgi?acc=GSE216075 | NCBI Gene Expression Omnibus, GSE216075 |
| Yang J | 2022 | Histone methylation mediates transgenerational modulations of sweet perception by high sugar diet [RNA-Seq] | https://www.ncbi.nlm.nih.gov/geo/query/acc.cgi?acc=GSE215756 | NCBI Gene Expression Omnibus, GSE215756 |

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
