## [Editor Report]

This study presents an important finding that high-sugar diet-induced behavioral changes can be transmitted to the offspring through the maternal germline. Using genetic and molecular biology approaches in the fruit fly model, the authors convincingly show that HSD has a transgenerational effect on PER, and that mothers fed an HSD produce progeny with globally elevated H3K27me3. The work will be of great interest to behaviorists and epigeneticists.

---

## [Decision Letter]

**Decision letter after peer review:**

Thank you for submitting your article "Exposure to high-sugar diet induces transgenerational changes in sweet sensitivity and feeding behavior via H3K27me3 reprogramming" for consideration by *eLife*. Your article has been reviewed by 3 peer reviewers, and the evaluation has been overseen by a Reviewing Editor and Michael Taffe as the Senior Editor. The following individual involved in the review of your submission has agreed to reveal their identity: Yan Li (Reviewer #3).

Essential revisions:

1) Please address the concerns about the sample size, variability, and statistics of the data presented.

2) Reviewers #2, and #3 suggested a list of control and additional experiments to strengthen the molecular and circuit mechanisms underlying the behavior. Please see the detailed suggestions listed below.

*Reviewer #1 (Recommendations for the authors):*

The phenotype of maternal influence is very interesting and worth further exploring. It's nice that knocking down Ez and Pcl expression during oogenesis eliminated the suppression of PER responses in HSD-F1 flies. Although it confirmed that the maternal H3K27me3 modification is critical for the transgenerational inheritance of behavioural change, the involvement of such modification on the paternal side is not ruled out.

It'll be great if the authors could confirm the effect of EED226 or A395 feeding on epigenomic modification status.

Page 19, last paragraph, 'it was unlikely that Cad was the sole gene mediated…' should be '… gene mediating…' or '…gene that mediated…'.

*Reviewer #3 (Recommendations for the authors):*

1) The procedures of HSD treatment in RNAi experiments are not clear enough. For instance, knocking-down E(z) or Pcl during oogenesis abolished the changes of PER in the F1 but not F0 (Figure 3-S1). I assume that Mata-tub-Gal4>UAS-xx-RNAi flies served as the F0 – please clarify. If so, this is a very important result to suggest that epigenetic modification during oogenesis is critical for passing on behavioral modification. To make this conclusion, genetic controls from both sides of the parents, as was performed in Figure 3, are needed.

Similarly, in Figure 3, the PER response needs to be examined in F0 as well. In addition, if nosNGT-Gal4>UAS-xx-RNAi flies served as F0, some of the F1 flies shared the same genotype with the F0 and some were not. Whether these F1 flies were examined separately? Or alternatively, nosNGT-Gal4>UAS-xx-RNAi flies were the F1, then it would lead to a different conclusion -the epigenetic modification in the F1 is critical for the behavioral change. It would be very helpful for better data interpretation if the treatment procedures were stated in more detail.

2) For the pharmacological experiments, EED226 and A395 were supplied to the F1, and both abolished the changes in PER response (Figure 4A-B). These results suggested that epigenetic modification is needed in the F1 – maybe only in F1 or in both F0 and F1. This could be tested by treating the F0 flies with these drugs. With a clear and comparable treatment timeline, the results obtained from pharmacological and genetic approaches could be discussed collectively.

3) For in vivo imaging of Gr5a+ neurons, the authors may look into the details of recording procedures in literature and pay attention to the way to stabilize the sucrose supply and other parts that may bring noises.

4) The authors show that cad expression is lower in the F1 and F2 flies, compared with normal diet (ND) flies (Figure 5E-F). How about the F0 flies? Whether knockdown of E(z) or Pcl could block the decrease in cad expression in the F0 and F1 flies?

5) Knocking-down E(z) and Pcl in Gr5a+ neurons have been shown to be able to block the behavioral change in PER response within the same generation. Whether it's sufficient to block the change in the next generation?

---

## [Author Response]

Essential revisions:Reviewer #1 (Recommendations for the authors):The phenotype of maternal influence is very interesting and worth further exploring. It's nice that knocking down Ez and Pcl expression during oogenesis eliminated the suppression of PER responses in HSD-F1 flies. Although it confirmed that the maternal H3K27me3 modification is critical for the transgenerational inheritance of behavioural change, the involvement of such modification on the paternal side is not ruled out.

We have now confirmed that HSD exposure suppressed PER in both male and female HSD-F0 flies (Figure 1. G, Figure 1—figure supplement 2. G). To test whether the transgenerational inheritance of behavioral changes was mediated by male and/or female germline, we performed HSD feeding in either female or male HSD-F0 flies and crossed them with ND-fed mates and found that reduction in PER was only seen in HSD-F1 progeny with HSD-fed female ancestor but not with HSD-fed male ancestor (Figure 1. I). We have also confirmed similar effect in HSD-F2 flies (Figure 1-figure supplement 2. I).

Therefore, we could conclude that transgenerational inheritance of PER reduction upon ancestral HSD exposure only transmitted through female but not male germline. We then moved on to investigate which and how epigenetic regulators were involved and identified H3K27me3 as the main driver (Figure 2). Therefore, it is not necessary to further test whether H3K27me3 was involved in the paternal side of transgenerational inheritance.

It'll be great if the authors could confirm the effect of EED226 or A395 feeding on epigenomic modification status.

We fed wild-type flies with H3K27me3 inhibitor EED226 for five days and used western blot to examine the levels of H3K27me3 of these flies. We found that compared to the control group fed with DMSO, flies fed with EED226 showed significantly decreased levels of H3K27me3, demonstrating the effectiveness of the inhibitor (Figure 4—figure supplement 1. A).

Page 19, last paragraph, 'it was unlikely that Cad was the sole gene mediated…' should be '… gene mediating…' or '…gene that mediated…'.

We have corrected the mistake in the manuscript.

Reviewer #3 (Recommendations for the authors):1) The procedures of HSD treatment in RNAi experiments are not clear enough. For instance, knocking-down E(z) or Pcl during oogenesis abolished the changes of PER in the F1 but not F0 (Figure 3-S1). I assume that Mata-tub-Gal4>UAS-xx-RNAi flies served as the F0 – please clarify. If so, this is a very important result to suggest that epigenetic modification during oogenesis is critical for passing on behavioral modification. To make this conclusion, genetic controls from both sides of the parents, as was performed in Figure 3, are needed.

We have modified schematic diagrams and added more detailed explanations in the manuscript in order to provide a clearer illustration of the experimental procedures for RNAi experiments and drug treatments (Figure 3. A, Figure 3—figure supplement 1. A, Figure 4. A, B and D). We have also added appropriate genetic controls from both sides of the parents (Figure 3. D-E, Figure 3—figure supplement 3. A-B).

Similarly, in Figure 3, the PER response needs to be examined in F0 as well. In addition, if nosNGT-Gal4>UAS-xx-RNAi flies served as F0, some of the F1 flies shared the same genotype with the F0 and some were not. Whether these F1 flies were examined separately? Or alternatively, nosNGT-Gal4>UAS-xx-RNAi flies were the F1, then it would lead to a different conclusion --the epigenetic modification in the F1 is critical for the behavioral change. It would be very helpful for better data interpretation if the treatment procedures were stated in more detail.

We have added clearer illustration of our experimental procedures. We also have added HSD-F0 data of nosNGT-Gal4>UAS-E(z)-RNAi flies (Figure 3-figure supplement 1. C). Notably, RNAi in early embryo stage did not rescue the decrease in PER of HSD-F0 flies PER because flies at the early embryo stage do not require feeding. By the time HSD-F0 flies developed into adults and started HSD feeding,

nosNGT-GAL4 was no longer active.

2) For the pharmacological experiments, EED226 and A395 were supplied to the F1, and both abolished the changes in PER response (Figure 4A-B). These results suggested that epigenetic modification is needed in the F1 – maybe only in F1 or in both F0 and F1. This could be tested by treating the F0 flies with these drugs. With a clear and comparable treatment timeline, the results obtained from pharmacological and genetic approaches could be discussed collectively.

We have added clearer illustrations of our experimental procedures (Figure 4. A-D).

We have conducted 5 days of pharmacological inhibition in adult HSD-F0 flies. The partial recovery of PER response in HSD-F0 after five days of drug treatment suggests that HSD exposure may still affect PER response of HSD-F0 flies through mechanisms independent of H3K27me3 (Figure 4. C).

We have also shown that EED226 feeding for five consecutive days restored PER responses of HSD-F1 flies and measured the sweetness sensitivity of their offspring HSD-F2 flies. We found that drug feeding in HSD-F1 parents could indeed block the decrease in PER response in the HSD-F2 offspring (Figure 4. D).

3) For in vivo imaging of Gr5a+ neurons, the authors may look into the details of recording procedures in literature and pay attention to the way to stabilize the sucrose supply and other parts that may bring noises.

As discussed above, we have improved the experimental procedures and achieved better data quality for the calcium imaging experiments (Figure 4. E). The conclusion remained unchanged.

4) The authors show that cad expression is lower in the F1 and F2 flies, compared with normal diet (ND) flies (Figure 5E-F). How about the F0 flies? Whether knockdown of E(z) or Pcl could block the decrease in cad expression in the F0 and F1 flies?

We have now measured the expression of Cad in the head of HSD-F0 flies and did not observe a significant decrease (Figure 5. F, left). We also measured the expression levels of Cad in HSD-F1 after knocking down E(z) and Pcl in HSD-F0 ovaries and found that the decrease in cad expression was indeed blocked (Figure 5. G).

5) Knocking-down E(z) and Pcl in Gr5a+ neurons have been shown to be able to block the behavioral change in PER response within the same generation. Whether it's sufficient to block the change in the next generation?

Indeed, knocking down E(z) in Gr5a^+^ neuron of both HSD-F0 and HSD-F1 flies could block the decrease in PER responses (Figure 4—figure supplement 1. B). But we did not conduct PER experiments in their progeny because transgenerational epigenetic inheritance should only transmit through the reproductive system rather than peripheral neurons. As we showed in Figure 3. D-E, knocking down E(z) and Pcl in the germline did modulate PER phenotype in the progeny.